# Cytoskeletal tension and Bazooka tune interface geometry to ensure fusion fidelity and sheet integrity during dorsal closure

Piyal Taru Das Gupta, Maithreyi Narasimha*

Department of Biological Sciences, Tata Institute of Fundamental Research, Mumbai, India

**Abstract** Epithelial fusion establishes continuity between the separated flanks of epithelial sheets. Despite its importance in creating resilient barriers, the mechanisms that ensure stable continuity and preserve morphological and molecular symmetry upon fusion remain unclear. Using the segmented embryonic epidermis whose flanks fuse during Drosophila dorsal closure, we demonstrate that epidermal flanks modulate cell numbers and geometry of their fusing fronts to achieve fusion fidelity. While fusing flanks become more matched for both parameters before fusion, differences persisting at fusion are corrected by modulating fusing front width within each segment to ensure alignment of segment boundaries. We show that fusing cell interfaces are remodelled from en-face contacts at fusion to an interlocking arrangement after fusion, and demonstrate that changes in interface length and geometry are dependent on the spatiotemporal regulation of cytoskeletal tension and Bazooka/Par3. Our work uncovers genetically constrained and mechanically triggered adaptive mechanisms contributing to fusion fidelity and epithelial continuity.

DOI: https://doi.org/10.7554/eLife.41091.001

**\*For correspondence:**
maithreyi@tifr.res.in;
mnarasimha.tifr@gmail.com

## Introduction

Epithelial fusion results in the formation of continuous epithelial sheets from the fusion of two epithelial flanks that are spatially separated by an intervening tissue or a gap, and accomplishes the closure of the neural tube, palate and epidermis in many organisms (*Jacinto et al., 2001*; *Kiehart et al., 2017*). In these contexts, fusion must not only ensure stable epithelial continuity but must also maintain morphological (size matching) and molecular (matching gene expression patterns) symmetry between the two fusing flanks. Defects in epithelial fusion morphogenesis can have disastrous consequences on embryonic viability resulting not only from the loss of tissue integrity but potentially also from the failure to maintain symmetry in gene expression patterns. An understanding of the molecular, cellular and physical principles that govern fusion fidelity and impart integrity to the newly formed seam is therefore essential from both fundamental and clinical perspectives.

Embryonic segmentation is a striking example of a molecular or genetic pre-pattern that also has a morphological correlate, and is a hallmark of both vertebrate and invertebrate body plans. Although the gene regulatory networks that govern segmentation were first identified in Drosophila over 40 years ago, the mechanisms that ensure segment continuity and the maintenance of symmetric segmentation gene expression patterns during fusion morphogenesis are remarkably poorly understood. Drosophila dorsal closure is a morphogenetic movement that relies on epithelial fusion and accomplishes the closure of a curved epithelial sheet, the embryonic epidermis, through the

meeting of their separated edges in the midline. The embryonic epidermis is morphologically and molecularly segmented along the anterior-posterior axis of the embryo. This segmentation is evident in the morphologically identifiable segmentation grooves and in the columnar expression patterns of segmentation genes during and after fusion. Epithelial fusion during dorsal closure must therefore not only ensure strong and stable epithelial continuity, but also the faithful matching of morphology and segmentation gene expression patterns.

Earlier studies on epithelial fusion driven morphogenetic movements including Drosophila dorsal closure (DC), ventral closure in *C. elegans* and neural tube closure in the chick have identified the cellular origins of forces that drive fusion. These studies have demonstrated roles for a) patterned and heterogeneous apical constriction in driving contraction of the intervening tissue, b) cell elongation and intercalation in driving the movement of the flanks and c) the supracellular actin cable and actin based lamellipodial and filopodial protrusions assembled in the leading edge cells in enabling proximity and recognition between fusing partners during fusion (*Eltsov et al., 2015*; *Haigo et al., 2003*; *Heller et al., 2014*; *Jacinto et al., 2000*; *Kiehart, 2015*; *Kiehart et al., 2000*; *Meghana et al., 2011*; *Millard and Martin, 2008*; *Narasimha and Brown, 2004*; *Nishimura et al., 2012*; *Peralta et al., 2008*; *Saravanan et al., 2013*; *Sokolow et al., 2012*; *Solon et al., 2009*; *Toyama et al., 2008*). Genetic and biophysical studies on Drosophila dorsal closure have revealed that the contraction of the amnioserosa to which the epidermal flanks are attached provides the major force that brings the flanks into close proximity (*Harden et al., 2002*; *Narasimha and Brown, 2004*; *Pasakarnis et al., 2016*; *Scuderi and Letsou, 2005*). Additionally, two forces originate in the epidermal flanks: a retarding force in the cells of the lateral epidermis and a driving force in the leading edge (or Dorsal Most Epidermal/DME) cells. The latter has been attributed to the apical supracellular actomyosin cable assembled in the DME cells of the two epithelial flanks at its fusing interfaces (*Kiehart, 1999*; *Kiehart et al., 2000*). Recent studies have argued that the actin cable is dispensable for driving closure, but demonstrate an effect on dorsal closure dynamics in its absence and suggest a role for the actin cable in facilitating scar less closure(*Ducuing and Vincent, 2016*; *Pasakarnis et al., 2016*). An actin cable is also assembled in wound, ventral and eyelid closure, but its requirement for the latter has been ruled out (*Heller et al., 2014*; *Raich et al., 1999*; *Rodriguez-Diaz et al., 2008*; *Williams-Masson et al., 1997*). Dynamic, short, actin based filopodia and lamellipodia that emanate from the fusing cell interfaces during dorsal closure are thought to contribute to forces that enable further proximity between the fusing flanks and to cell recognition and adhesion priming between fusing partners (*Eltsov et al., 2015*; *Jacinto et al., 2000*; *Millard and Martin, 2008*).

Surprisingly little is known about the mechanisms that ensure symmetry between the two fusing flanks and establish seamless epithelial continuity during Drosophila dorsal closure (*Kiehart et al., 2017*). Uncovering these mechanisms is of outstanding importance given the requirement of both stable epithelial continuity and geometric and molecular symmetry for the maintenance of integrity and the subsequent patterning of the structures engaged in fusion. Dorsal closure accomplishes the covering of the dorsal surface of the embryo by the cuticle producing epidermis and the alignment and registry of the embryonic segments Lb-A8. Pioneering studies that visualised labelled Drosophila embryonic epidermal segment compartments fusing during dorsal closure in real time, demonstrated their faithful pairing and alignment, and hinted at the requirement for cell pair matching between the fusing epidermal flanks (*Jacinto et al., 2000*; *Millard and Martin, 2008*). Studies based on electron microscopy images revealed filopodial interdigitations between fusing cell partners during Drosophila dorsal closure and suggested that such filopodial interdigitations must enable the fusing cell pairs to recognize each other and establish contact (*Jacinto et al., 2000*; *Eltsov et al., 2015*). An alternative possibility is that fusion fidelity is achieved through the spatiotemporal regulation of fusion, ensuring that only one pair of cells is proximate enough to fuse at any given time. Such a model would necessitate the spatiotemporal regulation of distance between the two flanks and of adhesion between the two fusing partners, one pair at a time. Whether this relies on the regulation of adhesion or contractility also remains unclear. Additionally, the nature and regulation of junctional changes that might impart mechanical integrity to and enable seamless continuity of the fused epithelial sheet remain unknown.

Using live confocal microscopy, quantitative morphodynamics and genetic perturbations, we qualitatively and quantitatively examine the progress of fusion of the embryonic epidermis during Drosophila dorsal closure. We investigate the basis of fusion fidelity and epithelial continuity and

uncover their dependence on changes in geometry and cell number in the fusing flanks. We further find that cytoskeletal tension and the polarity regulator Bazooka/Par 3 modulate fusing interface geometry and fusion front cell numbers to impart fidelity and enable seamless, stable epithelial continuity. Our findings reveal that fusion is not genetically pre-set to be accurate at the length scale of fusing cell pairs. Instead, we find that active, adaptive mechanisms that are spatially constrained to embryonic segment compartments and are mechanically triggered, contribute to fusion fidelity. Our work also uncovers for the first time, junction remodelling events that accompany epithelial fusion during dorsal closure.

## Results

### Chronology of segment fusion during dorsal closure

In a dorsal-up view of the embryo (used throughout in this study) with anterior to the left, the upper and lower arcs are the right and left arcs of the embryo. Fusion begins at the anterior and posterior canthi, proceeds inwards and results in the precise registry of the borders of embryonic segments Lb to A8 (*Figure 1A*). To follow the fusion of individual segment compartments, we labelled the posterior compartment (stripe) of each segment with UAS Actin GFP or UAS GFP driven by engrailed Gal4 (enGal4) in embryos that also carried a single copy of the ubi::ECadherin GFP transgene. This enabled the visualization of cell outlines in both compartments (the anterior compartment is referred to as the interstripe; *Figure 1A*, *Figure 1—figure supplement 1O–P* and *Figure 1—video 1*). At approximately 3 hr prior to the fusion ($-190$ mins) of the last segment (A3), the leading edge of the epidermis is just visible in a dorsal up view. About two hours prior to fusion ($-130$ mins), the segments Lb and A7 fuse. The remaining segments fuse after the onset of zippering ($-100$ mins) in quicker succession (*Figure 1B*, *Figure 1—figure supplement 1N*; n = 6 embryos).

### Measuring symmetry, fidelity and plasticity in fusion

Earlier work examined the final outcome of fusion of labelled segment compartments during dorsal closure and suggested that segment alignment (inferred from the precise registry of compartment boundaries) must involve cell matching in the two flanks (*Jacinto et al., 2000*; *Millard and Martin, 2008*). These studies did not address how symmetric the two flanks were, or whether, when and how is this symmetry achieved. To address these questions, we performed real-time, 3D confocal microscopy of dorsal closure in the embryos described above, and measured the lengths of the upper and lower arcs, the widths of each segment or segment compartment at the leading edge and the number of DME cells in each arc, each fusing segment or segment compartment during the course of dorsal closure. A population level analysis of cell numbers and widths of the compartments of all segments revealed that (i) the posterior compartment has fewer DME cells and is narrower than the anterior compartment and (ii) the posterior compartments of the last fusing segments (T3, A1-A4) exhibit the greatest plasticity, inferred from their heightened ability to modulate cell number and width (*Figure 1—figure supplement 1A–M*).

In order to obtain an estimate of fusion fidelity, and specifically to assess how symmetric or well matched the fusing right and left epidermal flanks or segments are, we performed a pairwise analysis of lengths and cell numbers between contralateral fusing partners (right and left flanks of the same segment) and measured the differences in these parameters between the fusing partners at multiple time-points during the course of fusion. For the analysis of fusion of segments or compartments, we defined two time points: one at the time of fusion of that segment ($t_F$) and another at an early time point ($t_E$). We also assessed whether fusing (contralateral) partners were more 'matched' than ipsilateral (non-fusing neighbours on the same side) compartments (*Figure 1C*; See Materials and methods).

### The evolution of high fidelity during dorsal closure

We first performed a pair-wise analysis of changes in DME cell number and arc length in the two flanks over the course of dorsal closure (*Figure 1D and I* and *Figure 1—figure supplement 2A–B*). Contrary to the expectations of a model in which fidelity is pre-set by invariant and equal cell numbers in the two fusing flanks, we found significant differences in both cell number and arc length between the fusing flanks at early time points. For both parameters, significant reductions in the

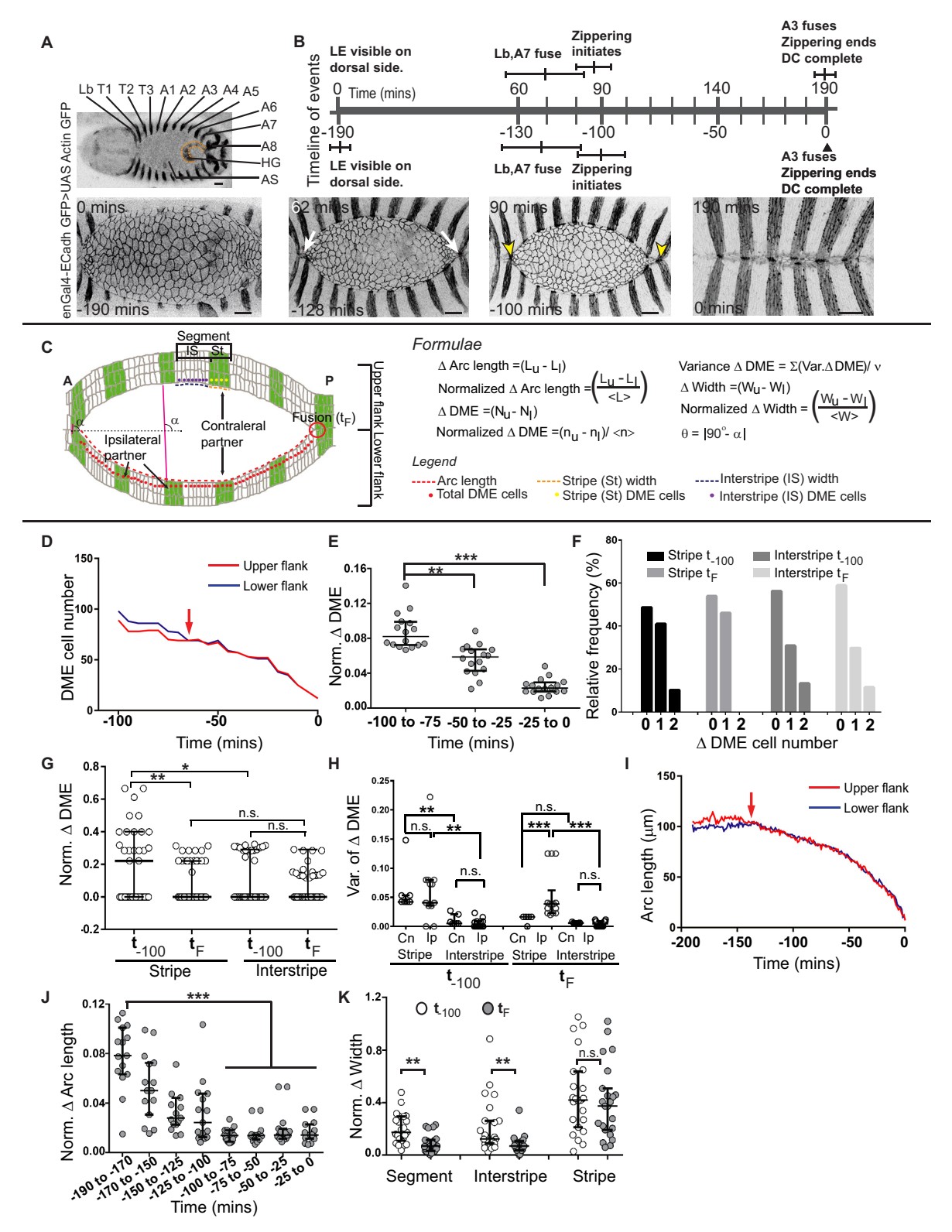

**Figure 1.** Chronology of segment fusion and pair-wise analysis of fidelity descriptors. (**A**) Top: Low magnification image of a Drosophila embryo at the onset of dorsal closure (DC) showing the segments (Lb-A8, posterior compartments/stripes are labelled with GFP) that fuse during dorsal closure (HG-hindgut and AS-amnioserosa). Bottom: Time-lapse images showing the progression of DC in one such embryo (white arrows show the fusion of the Lb and A7 stripes, yellow arrows mark the anterior and posterior canthi). (**B**) Prospective and retrospective time lines depicting the chronology of fusion

*Figure 1 continued on next page*

*Figure 1 continued*

during dorsal closure (times are mean ± sd, n = 5 embryos). (C) Epidermal landmarks and the descriptors used to assess fidelity during dorsal closure. DME - Dorsal Most Epidermal cells (also called Leading Edge cells; see Materials and methods). (D) DME cell number dynamics of the upper (red) and lower (blue) arc over the course of dorsal closure in a representative embryo in a retrospective time scale (0-completion of closure, red arrows denote time of equalization between the two arcs). (E) Normalized difference in DME cell number between the fusing arcs over the course of dorsal closure (median ± range, n = 16 embryos). (F) Frequency distribution of DME cell number differences between contralateral stripe and interstripe pairs at $t_{-100}$ and $t_F$ (n = 21 pairs of stripes/interstripes from seven embryos). (G) Normalized DME cell number difference in stripes and interstripes at $t_{-100}$ and $t_F$ (median ± range, n = 21 stripe/interstripe pairs from seven embryos). (H) Variance in DME cell number differences between contralateral and ipsilateral partner stripes and interstripes at $t_{-100}$ and $t_F$ (median ± range, n = 21 pairs of contralateral and 28 pairs of ipsilateral compartments from seven embryos). (I) Length dynamics of the upper (red) and lower (blue) arcs plotted over the course of DC in a representative embryo in a retrospective time scale (0-completion of closure, red arrows denote time of equalization). (J) Normalized difference in arc length between the fusing arcs over the course of dorsal closure (median ± range, n = 16 embryos). (K) Normalized width difference between contralateral partner segments and compartments at $t_{-100}$ and $t_F$ (median ± range, n = 21 pairs from seven embryos). Scale bar- 20 µm. (* - p<0.01, ** - p<0.001, *** - p<0.0001).

DOI: https://doi.org/10.7554/eLife.41091.002

The following video and figure supplements are available for figure 1:

**Figure supplement 1.** Segment specific differences in stripe and interstripe DME cell number and width.

DOI: https://doi.org/10.7554/eLife.41091.003

**Figure supplement 2.** Temporal changes in arc length and total DME cell number.

DOI: https://doi.org/10.7554/eLife.41091.004

**Figure supplement 3.** Cell addition to the leading edge coincides with stripe width equalization.

DOI: https://doi.org/10.7554/eLife.41091.005

**Figure 1—video 1.** Dynamics of native dorsal closure in embryos expressing UAS Actin GFP in engrailed stripes.

DOI: https://doi.org/10.7554/eLife.41091.006

**Figure 1—video 2.** Stripes, marked by UAS Actin GFP, showing different outcomes of cell rearrangement identified at the leading edge (from left to right: Type 1, Type 2, Type two and Type 3).

DOI: https://doi.org/10.7554/eLife.41091.007

disparity between the two fusing flanks were observed later. Reductions in the disparity in absolute and normalized arc length preceded the reduction in disparity in absolute and normalized cell number (100 and 50 minutes prior to closure for arc length and DME cell number respectively; *Figure 1E and J*; n = 16 embryos). These results demonstrate that the fusing flanks become more matched or symmetric with respect to DME cell numbers and arc lengths over time (*Figure 1—figure supplement 1Q*) and hint at the existence of distinct cellular mechanisms that accomplish changes in geometry and cell number to achieve symmetry and fusion fidelity. Surprisingly, mechanisms that contribute to equalization begin to operate well before fusion, suggesting they must operate at a distance.

## Differential contributions of segment compartments to equalization in cell number and geometry of the fusing fronts

To identify the cellular mechanisms that contribute to equalization in arc length and cell number between the two fusing flanks and to examine whether arc length and cell number equalization occurred uniformly along the anterior-posterior extent of the flank, we determined widths of and cell numbers in each pair of segment compartments that align and fuse during dorsal closure. For this analysis, we focused on the central segments that showed the maximum plasticity. Indeed, a higher proportion of central stripes showed a reduction in DME cell number disparity between fusing partners than the peripheral stripes or the central interstripes (*Figure 1—figure supplement 1E–J*; n = 6). On average, the stripes had only half as many DME cells as the interstripes (*Figure 1—figure supplement 1A–D*; n = 48 pairs).

Within the central stripes, DME cell number differences between contralateral fusing pairs ranged from 2 to 0 at $t_{-100}$ but reduced to 1 and 0 at $t_F$. Overall, the normalized disparity in cell number between contralateral stripes reduced by two-fold at fusion (*Figure 1G*). Significantly, while no stripe pairs with a difference in cell number of two were found at fusion, close to half of the stripe pairs examined showed a difference in cell number of one even at fusion (46.2%; n = 21 pairs; *Figure 1F and Figure 1—figure supplement 1F–G*). This demonstrates that a cell number difference of one between the fusing fronts of the posterior compartment at fusion is surprisingly common. In contrast, no significant reduction in cell number differences was observed between contralateral

interstripe pairs during the course of dorsal closure, and differences of one (in approximately 30%) or two cells (in 13%) persisted even at fusion (*Figure 1F* and *Figure 1—figure supplement 1I–J*). Indeed, the normalized difference in DME cell number of the stripe pair at fusion was similar to that exhibited by the interstripe at both time points (*Figure 1G*). This suggests that stripe pairs match cell numbers more effectively than the interstripe pairs at fusion.

In contrast, interstripes more effectively matched the widths of their fusing fronts (a 3 to 4-fold decrease in disparity at fusion) than the stripes, and contributed substantially to the equalisation of segment width (*Figure 1K*). Together, these results reveal that the anterior and posterior compartments of the central embryonic segments contribute different mechanisms to the temporal evolution of fidelity in fusion, and suggest that their ability to modulate the length and geometry of their fusing fronts may be genetically constrained.

## Differences in cell number are sensed and corrected within each segment compartment

The distinct contributions of the anterior and posterior compartments to fusion fidelity suggest that their differential gene expression patterns might contribute to it. We wanted to determine whether each segment compartment behaved autonomously to achieve symmetry in cell numbers. To address this, we examined whether ipsilateral compartments (adjacent stripes/interstripes on the same flank that do not fuse with each other) also exhibit a tendency to reduce disparity in cell numbers between them as we have demonstrated in contralateral partners (fusing partners on opposite flanks). For this, we compared the variance in DME cell number between ipsilateral and contralateral partners. Ipsilateral stripe pairs exhibited a significant increase in the variance in DME cell numbers at fusion compared to contralateral stripe pairs, but no significant differences in the variances in DME cell numbers were observed at early time points or between ipsilateral interstripes at fusion (*Figure 1H*; n = 21 contralateral pairs and 28 ipsilateral pairs). The observation that contralateral stripe pairs are more 'symmetric' with respect to DME cell numbers than ipsilateral stripes suggests that the segment compartment is the smallest unit within which differences in cell number are sensed and corrected. Together, our findings suggest that distinct genetically hardwired mechanisms operating within the posterior and anterior compartments must enable cellular rearrangements that facilitate changes in segment DME cell number.

## Cell addition to the leading edge does not reduce cell number disparity but may contribute to width equalization

To determine how differences in cell number between the fusing segment compartments are reduced during the course of dorsal closure, we visualised the evolution of changes in cell number by live confocal microscopy. We observed that all DME cell number changes in the stripe resulted from an increase in cell number in one or both flanks (*Figure 1—figure supplement 3A–D*). Two sources of cell addition were found: 'mixer cells' (MC) - DME cells that are plastic and cross from the anterior compartment and become integrated into the leading edge of the posterior compartment (and turn on engrailed as described [*Gettings et al., 2010*]) and 'posterior intercalating cells' (PIC; *Millard and Martin, 2008*) – cells that move from a row ventral to the DME cells in the posterior compartment and become incorporated in the leading edge (*Figure 1—video 2*). Cell addition occurred at different times in different stripes but reduced cell number disparity in only a third of the cases. Cell addition also occurred when there was no initial disparity, and in some cases cell addition increased the disparity (*Figure 1—figure supplement 3A–D,A'–D', E*). In every instance, cell number addition was followed by a transient or sustained equalization of compartment width (*Figure 1—figure supplement 3A"–D" and F*, n = 19 pairs). This suggests that cell addition may be primarily deployed to equalize the width of the posterior compartment.

Collectively, our results so far reveal that cell number matching occurs with less than perfect fidelity within segment compartments and point to the importance of matching fusing front geometry in the face of differences in cell number to ensure segment alignment. Our findings also suggest that mechanisms other than cell addition must contribute to width matching.

## Modulation of fusion front width is achieved by changes in fusing interface lengths within each segment compartment

Earlier work had demonstrated that the segment boundaries align at fusion (*Jacinto et al., 2000*; *Millard and Martin, 2008*). To determine the mechanisms by which the fusing fronts of the flanks become matched in length, we examined fusion in sqh::Utrophin GFP embryos in which the stripes were marked with nuclear RFP. This revealed that the dorsal-ventral aligned (D/V) interfaces of the DME cells at the posterior compartment (stripe) boundaries became aligned, with the anterior boundary more pronouncedly linear than the posterior boundary, through dynamic alterations in the length of the fusing anterior -posteriorly (A/P) oriented cellular interfaces despite differences in cell number in the fusing compartments (*Figure 2—figure supplement 1B* and Figure 6—video 3B). This suggests that anisotropic cell shape changes that principally modulate fusing interface lengths must contribute to the correction of cell number and width disparities that persist in segment compartments at fusion.

## Spatiotemporally regulated remodelling of cell junctions converts staggered en-face cellular contacts at the fusing front to an interlocking pattern after fusion

Earlier studies have attributed the accurate registry of segment compartments following fusion to filopodia-based mechanisms that ensure cell partner recognition and cell-to-cell adhesion (*Jacinto et al., 2000*; *Millard and Martin, 2008*). To identify the nature of the contacts that formed between fusing segments to ensure segment alignment and epithelial continuity, we examined the formation of the fusion front at high temporal and spatial resolution during and after fusion in embryos carrying ubi::ECadherin GFP. We also examined the nature of contacts in fixed enGal4 >UAS GFP embryos post-closure that were labelled with ECadherin and GFP. Our analysis revealed that despite a difference of one or two DME cells in the fusing segment compartments, the compartment boundaries aligned to form '+' shaped junctions. Such junctions formed between the straight, en-face contacts along the anterior-posterior axis in the dorsal midline (A/P oriented) and the aligned D/V interfaces of the cells at the compartment boundaries and ensured that the fusing fronts were now equal in width. Invariably, the shorter of the two fronts elongated till the boundaries were in alignment. In contrast, cells within segment compartments often contacted two cells on the opposite flank forming 'T'shaped junctions. Such junctions were formed between straight, en-face contacts formed along the anterior-posterior axis in the dorsal midline (A/P oriented) and the D/V interfaces of the cells within the compartment (Figures 2A and 5A,D, and *Figure 2—video 1*). We therefore wanted to examine whether these junctional configurations persist even after fusion.

Both kinds of contacts ('+' and 'T') resolved upon fusion to form an interlocking pattern, with the A/P interfaces becoming angled with respect to the dorsal midline to form tricellular junctions with the D/V interfaces. This pattern evolved from the en-face contacts through the gradual sliding and relaxation of straight, horizontal, A/P oriented interfaces, rendering them less taut, and their subsequent repositioning along the adjacent D/V interface which was shifted away from the dorsal midline (*Figure 2A*). Junction remodelling culminated in the formation of nearly evenly spaced, 'Y'shaped tricellular junctions between two adjacent A/P interfaces and the D/V interface and obliterated the midline seam approximately 15 minutes after fusion. This remodelling was associated with a small decrease in ECadherin GFP intensity in the fusing interfaces as they interlocked (*Figure 2A* and *Figure 2—video 1*). These findings reveal that fusion does not always culminate in the accurate matching of cell pairs and suggest that the final conformation of junctions (interlocked, tricellular) confers stability and mechanical strength to the newly formed seam, rendering it less susceptible to tears. Thus, the geometric and molecular remodelling of the newly formed ECadherin-based contacts that includes changes in interface length, position and possibly also tension accomplishes strong and stable epithelial continuity upon fusion (*Figure 2G,H*). To identify the mechanisms that contribute to the remodelling of contacts between the two fusing flanks, we examined the spatiotemporal organization of the actomyosin cytoskeleton and of regulators of adhesion.

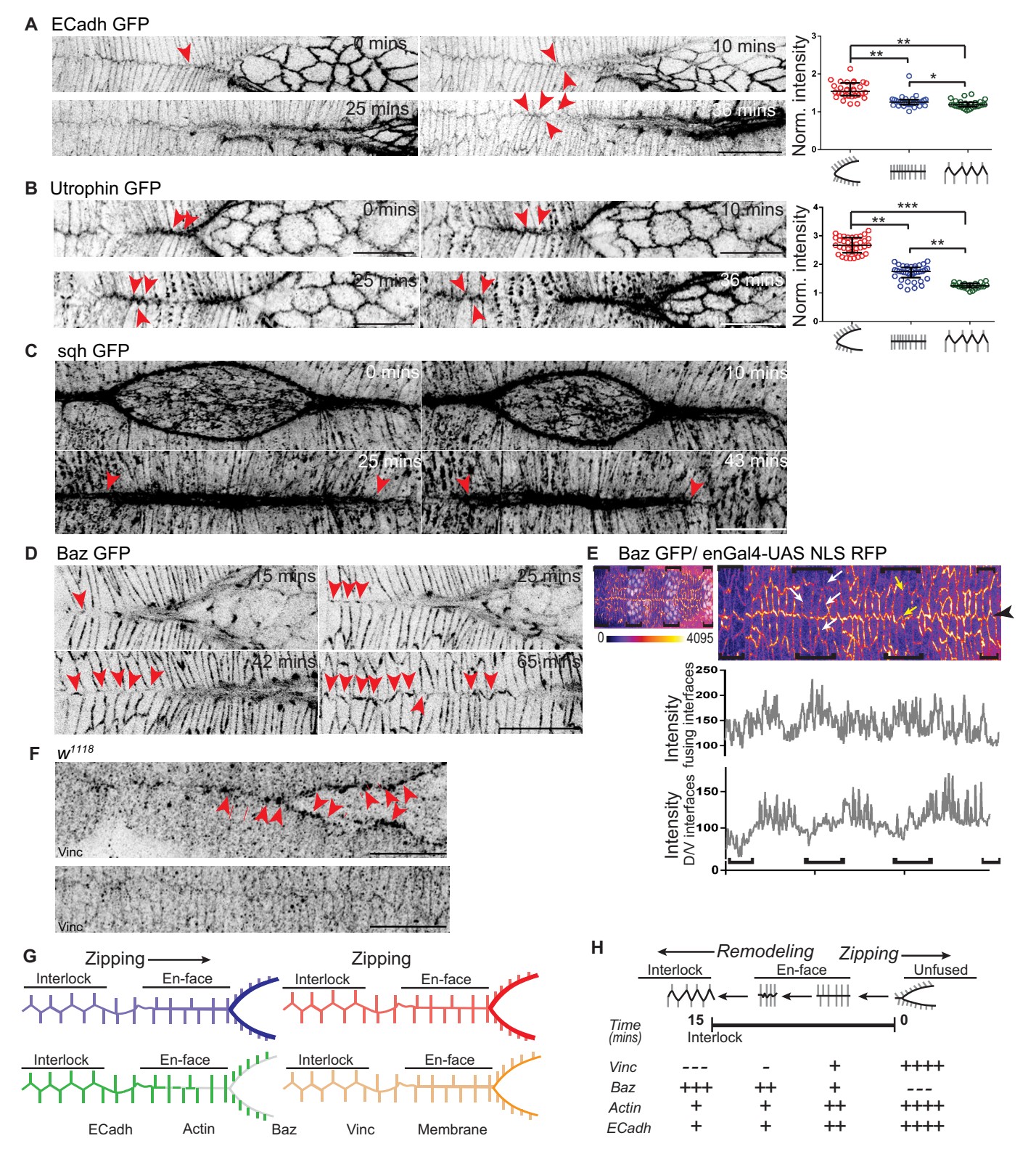

**Figure 2.** Dynamic changes in the geometry and molecular composition of the fusing interface. (**A, B**) High resolution time lapse images of embryos expressing ECadh GFP (A; n = 8 embryos) and Utrophin GFP (B; n = 9 embryos) showing the progressive remodelling of en-face contacts at fusion to an interlocking pattern after fusion at the canthus (red arrowheads). The cluster plots alongside A and B show the normalized intensity of ECadh GFP and Utrophin GFP respectively at the fusing interfaces before, during and after fusion (circles indicate individual data points; black lines indicate

*Figure 2 continued on next page*

*Figure 2 continued*

median ± range; n = 30 sets of 10 interfaces (from five embryos) for ECadh GFP and 37 sets of 5 interfaces each (from six embryos) for Utrophin GFP; (*- p<0.03, **- p<0.001, ***- p<0.0001)). (C) High resolution time lapse images of embryos expressing sqh GFP (C; n = 3 embryos) showing the intensity receding at the midline some interfaces away from the canthus as the interfaces are interlocked (red arrowheads). (D) High resolution time lapse images of embryos expressing Bazooka GFP (Baz GFP; n = 11 embryos) showing the recruitment of Bazooka upon fusion and its progressive enrichment with interlocking at the fusing front (red arrowheads). (E) Top: Embryo expressing Baz GFP (heat map) and enGal4-UAS NLS RFP (white nuclei) showing differential expression of Bazooka along the D/V interfaces (white arrows) and non-leading edge A/P interfaces (yellow arrows) in stripes (black brackets) compared to interstripes. Bottom: Line intensity profiles of the same image at the fusing interfaces and at the D/V interfaces of DME cells. (F) Fixed preparations showing localization of Vinculin (Vinc) in the DME cells during (top) and after (bottom) fusion in wildtype embryos (n = 8 embryos; red arrowheads show Vinculin enrichment at unfused and en-face contacts). (G) Schematic representation of the dynamic changes in the localization of molecules at the fusing interfaces. (H) Pictorial summary of the dynamics of the molecular composition and its correlation with dynamic modulations of interface geometry. Scale bar- 20 µm.

DOI: https://doi.org/10.7554/eLife.41091.008

The following video and figure supplement are available for figure 2:

**Figure supplement 1.** Dynamics of Bazooka and Vinculin localization, interface geometry and actin cytoskeleton organization at the leading edge.
DOI: https://doi.org/10.7554/eLife.41091.009

**Figure 2—video 1.** Remodelling of fusing DME interfaces from en-face contacts to an interlocking pattern (yellow arrowheads) visualized in embryos expressing ECadh GFP.
DOI: https://doi.org/10.7554/eLife.41091.010

**Figure 2—video 2.** Actin dynamics during the remodelling of fusing interfaces from en-face contacts to an interlocking pattern (yellow arrowheads) visualized in embryos expressing Utrophin GFP.
DOI: https://doi.org/10.7554/eLife.41091.011

**Figure 2—video 3.** Myosin dynamics during the remodelling of fusing interfaces from en-face contacts to an interlocking pattern (yellow arrowheads) visualized in sqh GFP embryos.
DOI: https://doi.org/10.7554/eLife.41091.012

**Figure 2—video 4.** Dynamic localization of Bazooka at the fusing interfaces during native dorsal closure in embryos expressing genomic Baz GFP.
DOI: https://doi.org/10.7554/eLife.41091.013

## Spatiotemporal regulation of actomyosin organisation: clearance of actin from fused interfaces

The transition in interface geometry that we described above is reminiscent of the changes that accompany cell intercalation during germ band extension in *Drosophila*, in which the conversion from Type I to Type II junctions involves the resolution of one or more four-point vertices by junction growth in the perpendicular direction. This remodelling restores the formation of the more stable three-point vertices (*Bertet et al., 2004*; *Blankenship et al., 2006*; *Simões et al., 2010*). Regulation of junction length during growth involves both passive relaxation and active cytoskeletal changes (*Collinet et al., 2015*; *Hara et al., 2016*). We therefore examined cytoskeletal organization and dynamics at the fusing interfaces during fusion. Changes in the levels of actin (visualised using Utrophin GFP) and myosin (visualised using sqh GFP) accompanied interface remodelling. Specifically, while both were enriched in unfused interfaces and en-face contacts, their levels reduced significantly in angled, bent and interlocked interfaces (*Figure 2B,C* and *Figure 2—video 2, 3*). Live imaging with Utrophin GFP revealed that actin is 'cleared' from the interlocking interfaces (*Figures 2B* and *6C*, *Figure 2—video 2* and Figure 6—video 3A). All three proteins (actin, myosin and ECadherin) were specifically regulated in the fusing interfaces (intensities in *Figure 2A–B* are normalized to that of D/V oriented interfaces in the same cells). The D/V oriented interfaces of DME cells showed no change in the levels of actin or ECadherin (*Figure 2—figure supplement 1E–F*). These results suggest that the spatiotemporally regulated, planar polarized distribution dynamics of the actomyosin cytoskeleton and possibly also of ECadherin may contribute to the resolution of the fusing front to a seamless, interlocked geometry.

The polarity regulator Par3/Bazooka is known to influence both actomyosin organization and cadherin dependent adhesion and guide junction remodelling. Indeed, the conversion from Type I to Type II junctions during cell intercalation is associated with the appearance of Bazooka. The mutual exclusion of myosin and Bazooka enables junction shortening in the Type I interfaces that do not contain Bazooka. Conversely, Bazooka-enriched interfaces lengthen (*Bertet et al., 2004*; *Blankenship et al., 2006*; *Simões et al., 2010*). We therefore wished to examine whether Bazooka may also be spatiotemporally regulated at the fusing interfaces during the fusion of the epidermis.

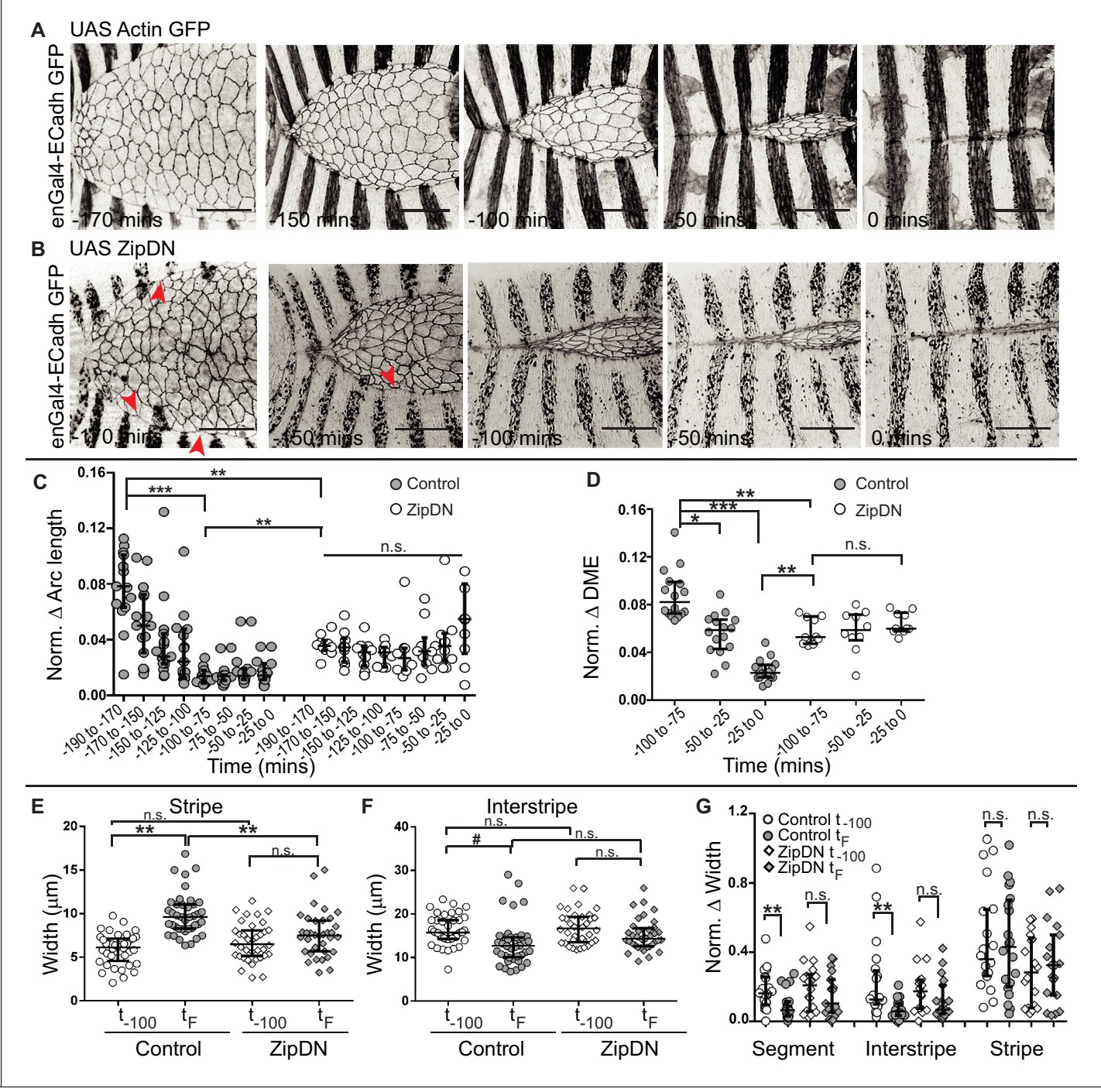

**Figure 3.** Tissue tension entrains the temporal evolution of fidelity in epithelial fusion. (**A, B**) Time-lapse images of dorsal closure in control embryos (**A**), and in embryos expressing UAS ZipDN driven by enGal4 (**B**). Red arrowheads denote scalloped leading edge. (**C, D**) Normalized arc length (**C**) and DME cell number (**D**) differences in control and in UAS ZipDN embryos (median ± range, n = 16 for control, n = 10 for UAS ZipDN). (**E–F**) Absolute widths of central stripes (**E**) and interstripes (**F**) of control and UAS ZipDN embryos at $t_{-100}$ mins and at $t_F$ (median ± range, n = 40 stripes/interstripes from 10 embryos). (**G**) Normalized difference in width of contralateral partners at $t_{-100}$ and $t_F$ (median ± range, n = 20 pairs from segments A2-A3 from 10 embryos). (# - p<0.05, *- p<0.01, **- p<0.001, ***p<0.0001). Scale bar- 40 µm.

DOI: https://doi.org/10.7554/eLife.41091.014

The following video and figure supplement are available for figure 3:

**Figure supplement 1.** The influence of tension in the epidermis on the dynamics and fidelity of fusion during dorsal closure.

DOI: https://doi.org/10.7554/eLife.41091.015

*Figure 3 continued on next page*

*Figure 3 continued*
**Figure 3—video 1.** Dynamics of dorsal closure in an embryo expressing ZipDN in stripes.
DOI: https://doi.org/10.7554/eLife.41091.016

## Spatiotemporal regulation of Bazooka recruitment to the fusing interfaces

To determine whether Bazooka may also enable junction remodelling in the context of 'reepithelialisation' or the formation of new contacts during epithelial fusion, we examined the distribution of Bazooka during and after fusion. Live imaging of genomic Bazooka GFP (Baz GFP) embryos revealed striking spatiotemporal changes in Bazooka distribution. Bazooka was absent from pre-fusion DME interfaces that were rich in actin and myosin but became recruited to these interfaces as the en-face contacts began to form. Its intensity at these interfaces increased as the contacts began to bend and reposition (11.2 ± 0.3 mins after initial contact; n = 71 interfaces), and peaked when the interfaces became configured to create an interlocking cell arrangement (*Figure 2D* and *Figure 2—figure supplement 1A*; n = 11 embryos and *Figure 2—video 4*). Bazooka colocalised with ECadherin containing cell junctions at the fusing interfaces (*Figure 5—figure supplement 1G*). These results reveal the complementary patterns of distribution of actomyosin and Bazooka at the fusing interfaces. While the intensities of Bazooka GFP in the fusing interfaces of both the stripe and the interstripe were comparable, Bazooka distribution in other A/P or D/V oriented interfaces was lower in the stripe than in the interstripe (*Figure 2E*). This differential distribution within segment compartments may regulate compartment specific differences in the cellular behaviours that contribute to fusion fidelity.

## Spatiotemporal regulation of cytoskeletal tension: Vinculin clearance from fused interfaces

We wished to determine whether the junctions that Bazooka is newly recruited to might also be more relaxed. For this, we examined the distribution of Vinculin, an adherens junction and focal adhesion protein whose recruitment to the cytoplasmic face of the adherens junction is dependent on force or tension dependent changes in the conformation of the junctional protein α−Catenin (*Hara et al., 2016*; *Seddiki et al., 2018*; *Yonemura et al., 2010*). We examined Vinculin distribution using an antibody that recognizes Vinculin and in embryos carrying a Vinculin GFP transgene. In sharp contrast to Bazooka, Vinculin in the DME cells was enriched in unfused interfaces and in en-face contacts where its highest levels were found in dot like structures. After fusion, the dots were no longer visible at the apical membrane, and lower levels of Vinculin were detectable along the length of the interface (*Figure 2F*). Live imaging with Vinculin GFP revealed that while the dots observed pre-fusion persist for some time after fusion, they appear to have repositioned basally. This was particularly evident in orthogonal sections from such embryos and in fixed embryos carrying Vinculin GFP (*Figure 2—figure supplement 1C–D*). This reciprocity suggests that Bazooka enrichment at the newly formed cell junctions is associated with reduced interfacial tension. These findings establish the reciprocal dynamics of actomyosin/interfacial tension and Bazooka at fusing DME cell interfaces (*Figure 2G–H*). Vinculin GFP also showed a striped expression pattern in the epidermis (*Figure 2—figure supplement 1C–D*), suggesting that tension within segment compartments might be differentially regulated.

To determine whether the spatiotemporal regulation of interfacial tension and Bazooka contribute to fusion fidelity and seamless epithelial continuity, we used genetic perturbations that modulate cytoskeletal tension or Bazooka levels and distribution within the posterior compartment and examined their effects on fusion fidelity, epithelial continuity and the mechanisms that contribute to it.

## Cytoskeletal tension entrains the temporal evolution of fidelity and modulates cell behaviour

The observations described above demonstrate that the supracellular actomyosin cable that assembles along the fusing DME cell interfaces/arcs and provides a force for closure prior to fusion, disappears after fusion. This suggests that there is a progressive relaxation of the fusing interfaces after fusion. We therefore examined the role of tension generated by the cytoskeleton on fusion fidelity and epithelial continuity. For this, we measured the fidelity descriptors described earlier (DME cell

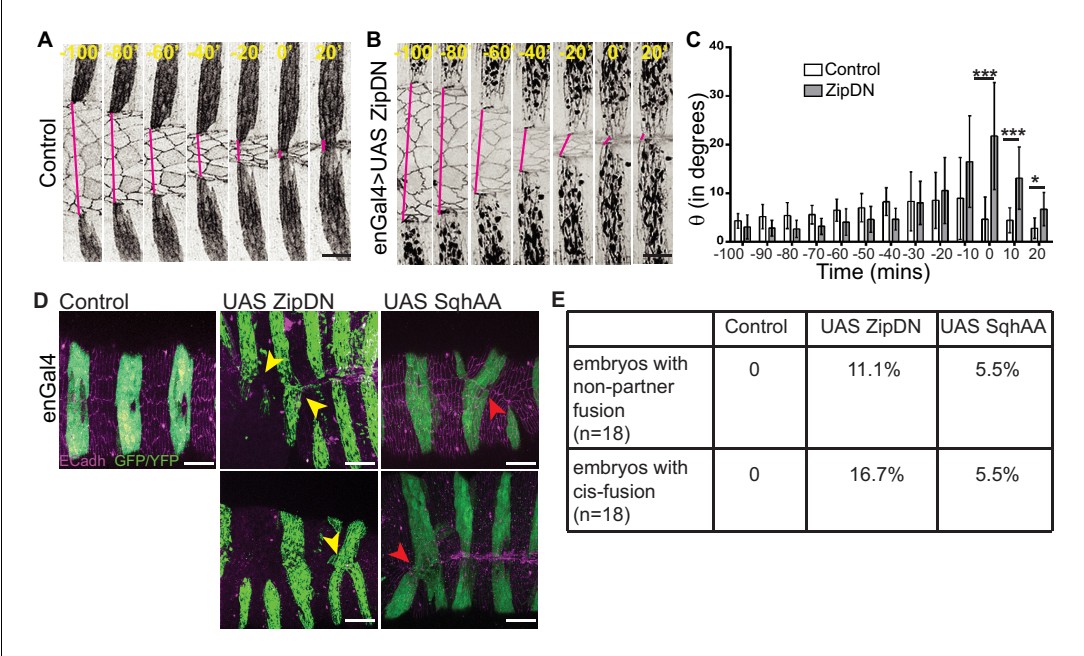

**Figure 4.** Tissue tension modulates the alignment and matching of contralateral partner stripes at the midline. (A, B) Time series images showing the positions of the anterior boundaries (red lines) of a pair of contralateral partner stripes from segment A3 in control embryos (A) and in embryos expressing UAS ZipDN (B). Scale bar-10 μm. (C) Temporal evolution of alignment of contralateral partner stripes in control and UAS ZipDN (mean ± sd, n = 10 pairs from 10 embryos, *- p<0.01, ** -p < 0.001, *** -p < 0.0001). (D) UAS ZipDN and UAS SqhAA expressing embryos showing non-partner fusion (yellow arrowheads) and cis-fusion (red arrowheads). Scale bar- 20 μm. (E) Frequency distribution of fusion defects in control embryos, and in embryos overexpressing UAS ZipDN or UAS SqhAA.

DOI: https://doi.org/10.7554/eLife.41091.017

numbers and widths of the fusing segment compartment fronts as well as the alignment of segment boundaries) in embryos expressing a contraction defective form of myosin (ZipDN) previously shown to function in a dominant negative manner, in the stripes (*Dawes-Hoang et al., 2005*; *Saravanan et al., 2013*) (*Figure 3A–B*, *Figure 3—figure supplement 1A* and *Figure 3—video 1*). We perturbed the stripes rather than just the DME cells since the cell rearrangements we had documented include cells that are not initially at the leading edge. In such embryos, both the fusion of stripe pairs and dorsal closure was delayed (*Figure 3—figure supplement 1B–D*). Additionally, the arc lengths and DME cell numbers were more variable than controls during early closure (*Figure 3—figure supplement 1E–F*; n = 16 and 10 for control and UAS ZipDN respectively). Despite this, both parameters reduced in mutant embryos with the progression of closure, albeit with different kinetics compared to control embryos (*Figure 3—figure supplement 1E–F*). Within individual embryos, neither parameter exhibited the temporal evolution that was observed between contralateral partners in control embryos (*Figure 3C–D*). Instead, the mutant embryos appeared invariant over time compared to control embryos, and the maximum reduction in disparity was also lower. The stripe and interstripe widths did not show any change between early ($t_{-100}$) and late ($t_F$) time points (*Figure 3E–F*; n = 40 stripes/interstripes). Additionally, unlike controls, the interstripes in these embryos also did not show a significant reduction in width disparity (*Figure 3G*; n = 20 pairs). Thus, cytoskeletal tension generated by actomyosin interactions entrains the temporal progression to high fusion fidelity. ZipDN expressing stripes were also significantly less well aligned at and after fusion compared to controls (*Figure 4C*; n = 10 pairs each for control and Zip DN). In a fraction of such embryos and in a smaller percentage of embryos expressing a phosphorylation defective myosin regulatory light chain (SqhAA, *Jordan and Karess, 1997*), fusions with ipsilateral or non-partner contralateral stripes were observed (*Figure 4D–E*; n = 10 pairs). These results strongly suggest that cytoskeletal tension exerts both a restraining and a permissive force that guides the temporal evolution of fidelity during epithelial fusion.

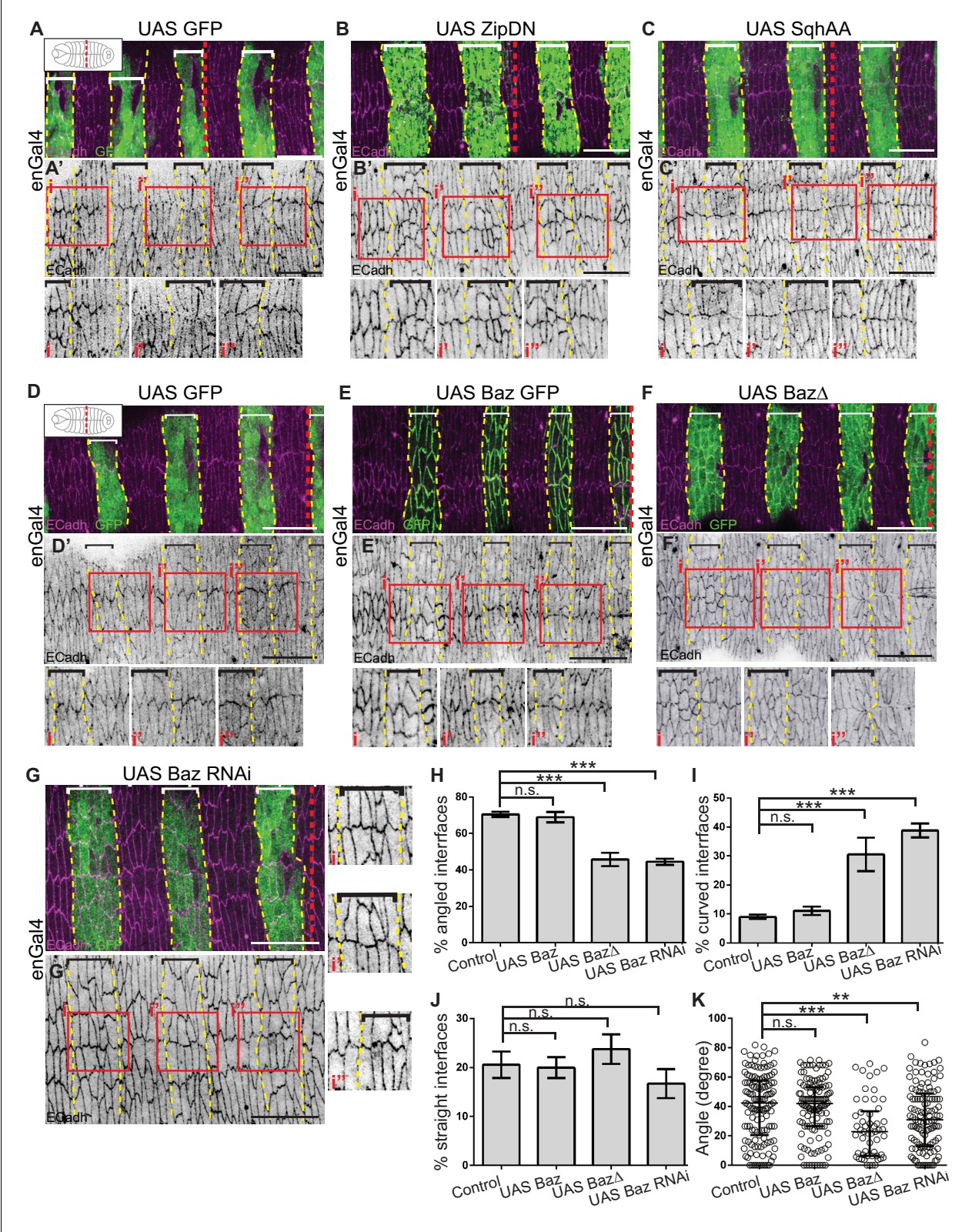

**Figure 5.** Cytoskeletal tension and Bazooka influence interface remodelling, cell shape and epithelial organisation during fusion. (**A–G**) Interface geometries of fusing DME cells and epithelial organisation in embryos expressing UAS GFP (control; A, D n = 8, 14 embryos), UAS ZipDN (B, n = 14 embryos), UAS SqhAA (C, n = 15 embryos), UAS Baz GFP (E, n = 9 embryos), UAS BazΔ (F, n = 10 embryos) and UAS Baz RNAi (G, n = 14 embryos) after fusion stained for ECadh (magenta) and GFP (marking the engrailed compartment, green). A'- G' are single channel images showing ECadh

*Figure 5 continued on next page*

*Figure 5 continued*

localization. Dashed yellow lines mark the boundaries of expressing stripes (bracketed). The dashed red lines in A-G mark the midpoint along the A/P axis (as shown in cartoons inset in A, D). The red boxes in A'-G' are magnified in i, i' and i''. Scale bar- 20 µm. (**H–K**) Frequency distribution of interface geometries (**H–J**) and distribution of interface angles (**K**) of stripe DME cells at the leading edge in embryos expressing UAS GFP (control, n = 135 interfaces, 11 embryos), UAS Baz GFP (n = 123 interfaces, nine embryos), UAS BazΔ (n = 54 interfaces, seven embryos) and UAS Baz RNAi (n = 126 interfaces, 11 embryos) driven by enGal4. (** -p < 0.01, ***-p < 0.0002).

DOI: https://doi.org/10.7554/eLife.41091.018

The following figure supplement is available for figure 5:

**Figure supplement 1.** The influence of cytoskeletal tension and Bazooka on interface geometry, epithelial organization and epithelial integrity.

DOI: https://doi.org/10.7554/eLife.41091.019

## Cytoskeletal tension modulates changes in interface geometry during contact remodelling

We then examined whether cytoskeletal tension also influences junction remodelling after fusion. Expression of either ZipDN or SqhAA in stripes resulted in the poor resolution of junctions to an interlocking pattern after fusion (in 95% of stripes in UAS ZipDN, n = 80 stripes and 96.5% stripes in UAS SqhAA, n = 102 stripes; *Figure 5A–C*). Surprisingly both perturbations also affected junction remodelling in the interstripe (81.3% of interstripes in UAS ZipDN, and 88.5% interstripes in UAS SqhAA). The two perturbations however produced qualitatively different effects on epithelial organization within the stripe. Whereas the expression of ZipDN produced pronounced epithelial disorganization at the dorsal midline, the expression of SqhAA had a more uniform effect on all interfaces resulting in an unobliterated midline seam, suggesting a block in junction repositioning (*Figure 5A–C*). Consistent with this, the frequency distribution of angled A/P interfaces in DME cells from stripes expressing SqhAA was significantly lower and that of straight A/P interfaces significantly higher than in control stripes (*Figure 5—figure supplement 1E*). The angles subtended by the interfaces were also lower than that of controls (*Figure 5—figure supplement 1F*). A similar analysis could not be done on UAS ZipDN embryos on account of the greater epithelial disorganization at the midline. These results suggest that spatiotemporally regulated changes in cytoskeletal tension in the fusing DME interfaces drive timely interface remodelling. These experiments do not however enable us to separately examine the influence of tension generated by the actomyosin cytoskeleton at the fusing interfaces, compartment boundaries (*Monier et al., 2010*), or elsewhere within the stripe.

## Modulation of cytoskeletal tension in the DME cells influences the straightness of the fusion seam

To determine whether tension in the whole stripe or just in the DME cells governed its effects on fusion fidelity, we also expressed ZipDN, SqhAA and SqhEE (a phosphomimetic version of the myosin regulatory light chain) exclusively in the DME cells using a leading edge cell specific Gal4 (LEGal4) that is expressed patchily. All three perturbations influenced the straightness of the midline seam formed upon fusion as well as its obliteration after fusion. Specifically, expression of both Zip DN and SqhAA rendered the seam (the blue line drawn by joining the interfaces that contribute to it) less straight (bent or wavy) with respect the anterior posterior axis or dorsal midline (the red line in *Figure 5—figure supplement 1A,A'–C,C'*). In contrast, the seam in embryos expressing SqhEE was straight rather than jagged (*Figure 5—figure supplement 1D,D'*). All three perturbations produced patches of defective interlocking (*Figure 5—figure supplement 1A", A"', D', D"'*), suggesting that the spatiotemporally regulated balance of forces between the A/P and D/V oriented interfaces is critical for orderly interlocking and DME cell spacing both along the D/V and A/P axes.

## Bazooka modulates interface remodelling during fusion

As mentioned earlier, the polarity regulator Par3/Bazooka has been shown to influence both actomyosin organization and cadherin dependent adhesion, and to guide the conversion from Type I to Type II junctions during cell intercalation (*Bertet et al., 2004*; *Blankenship et al., 2006*; *Simões et al., 2010*). To examine whether Bazooka might also influence interface remodelling during the formation of new contacts during fusion (reepithelialisation), we expressed GFP tagged full length Bazooka, Bazooka RNAi and a GFP tagged C terminally truncated (Δ969–1464/BazΔ) Bazooka

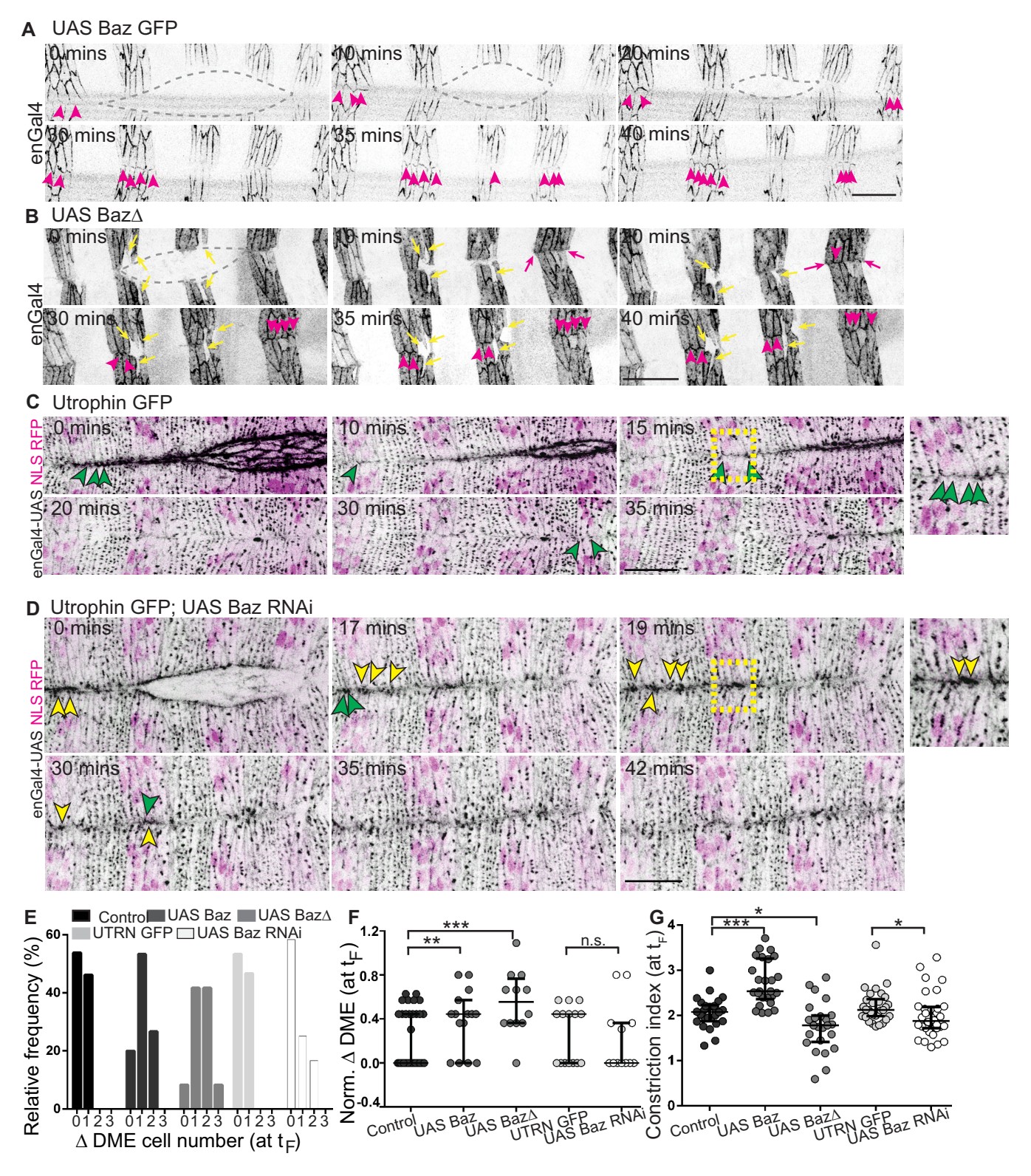

**Figure 6.** Bazooka localizes at newly formed junctions and influences interface remodelling and fusion fidelity. (**A–D**) Time-lapse images of embryos expressing UAS Baz GFP (A, n = 6 embryos), UAS BazΔ (B, n = 6 embryos), UAS NLS RFP alone (C, n = 6 embryos) or UAS Baz RNAi and UAS NLS RFP (D, n = 7 embryos) in the stripes. Embryos in C, D also express Utrophin GFP. Grey broken lines in A, B mark the dorsal opening. Magenta arrowheads in A, B show Bazooka localization at the newly formed contacts between epithelial cells from the opposing flanks; magenta arrows (in B) indicate stripe

*Figure 6 continued on next page*

*Figure 6 continued*

overhangs after fusion and yellow arrows (in B) indicate improperly positioned mixer cells along the A/P axis. Green and yellow arrowheads in C, D indicate respectively, the dynamics of interface remodelling and of Utrophin GFP, magnified in the insets provided alongside (region magnified is marked by the dashed yellow squares). (E, F) Absolute (E) and normalized differences (F) in DME cell number between contralateral partner stripes at fusion ($t_F$) of UAS Actin GFP (control), UAS Baz GFP, UAS BazΔ, Utrophin GFP/enGal4-UAS NLS RFP (UTRN GFP, control) and of UAS Baz RNAi (with Utrophin GFP/enGal4-UAS NLS RFP) expressing embryos (n = 30 pairs of stripes from 10 embryos for control, 15 pairs of stripes from five embryos for UAS Baz GFP, 12 pairs of stripes from four embryos for UAS BazΔ, 15 pairs of stripes from five embryos for UTRN GFP and 15 pairs of stripes from six embryos for UAS Baz RNAi). (G) Constriction index of stripes at fusion in control (enGal4 >UAS Actin GFP, n = 26 stripes from four embryos), enGal4 >UAS Baz GFP (n = 26 stripes from four embryos), enGal4 >UAS BazΔ (n = 24 stripes from four embryos), UTRN GFP (n = 30 stripes from five embryos) and enGal4 >UAS Baz RNAi (n = 30 stripes from five embryos). (median ±range; *- p<0.01, **- p<0.001, ***- p<0.0001). Scale bar- 20 μm.
DOI: https://doi.org/10.7554/eLife.41091.020

The following video and figure supplement are available for figure 6:

**Figure supplement 1.** Dynamics of Bazooka recruitment and its influence on interface geometry.
DOI: https://doi.org/10.7554/eLife.41091.021
**Figure 6—video 1.** Dynamics of Bazooka localisation, stripe fusion and junction resolution in stripes expressing Baz GFP.
DOI: https://doi.org/10.7554/eLife.41091.022
**Figure 6—video 2.** Dynamics of Bazooka localisation, stripe fusion and junction remodelling in stripes expressing BazΔ.
DOI: https://doi.org/10.7554/eLife.41091.023
**Figure 6—video 3.** Dynamics of stripe fusion visualised in embryos expressing Utrophin GFP and enGal4-UAS NLS RFP.
DOI: https://doi.org/10.7554/eLife.41091.024
**Figure 6—video 4.** Dynamics of stripe fusion, junction resolution and actin organization at the leading edge in embryos expressing Baz RNAi in stripes and Utrophin GFP.
DOI: https://doi.org/10.7554/eLife.41091.025
**Figure 6—video 5.** Laser ablation of a Baz GFP overexpressing stripe prior to (left two panels) and after fusion (right).
DOI: https://doi.org/10.7554/eLife.41091.026

transgene in the stripes. The latter was previously demonstrated to exhibit dominant negative effects in the context of neuronal polarity and shown to prevent its membrane recruitment by interfering with its interactions with phosphoinositides (*Krahn et al., 2010*; *Shi et al., 2003*). To examine the effects of Bazooka RNAi in the stripes, we used UAS GFP (in fixed preparations) or UAS NLS RFP (for live imaging) to label the stripe, and sqh::Utrophin GFP to enable the visualization of changes in cell and interface geometry and cytoskeletal organization in real time. We chose Utrophin GFP (rather than ECadherin GFP or sqh GFP) since Bazooka has been shown to regulate both ECadherin and myosin albeit in different contexts. It was therefore possible that ECadherin GFP or sqh GFP may modify the phenotype resulting from Bazooka knockdown.

We first analysed the effect of Bazooka perturbations on interface remodelling in fixed preparations. The expression of Baz RNAi or BazΔ in stripes resulted in a significant increase of fused interfaces that were curved/squiggly compared to controls, and a significant decrease in angled interfaces. In contrast Bazooka overexpression did not significantly alter the distribution of interface types (*Figure 5D–J*). Further, the average angle of the fusing interfaces was significantly reduced upon expressing Baz RNAi (32.3 ± 2°) and BazΔ (25.1 ± 2.7°) compared to the control (39.7 ± 2°) and Baz overexpression (39.1 ± 1.7°) (*Figure 5K*). We also examined interface morphologies in zygotic *baz*[4] mutant embryos. Dorsal closure fails in the majority of such mutants precluding a detailed analysis. However, in the few embryos that do progress through dorsal closure, the seam is characterized by straight interfaces (*Figure 5—figure supplement 1J*). These results suggest that Bazooka function modulates interface morphology to facilitate interlocking.

We then examined fusing interface morphologies in the three perturbations in real time. For the full length and truncated Bazooka transgenes, we used their GFP tags to visualize interface morphologies at the fusing DME interfaces. As mentioned above, we used sqh::Utrophin GFP to visualize interface morphologies in Baz RNAi. Live imaging of fusion in stripes overexpressing full length Bazooka recapitulated the recruitment dynamics seen in genomic Baz GFP embryos. Overexpressed Bazooka was recruited to enface contacts of DME cells and subsequently became enriched (15–20 minutes after the initiation of fusion) in interlocking interfaces (*Figure 6A* and *Figure 6—video 1A-B*). This suggests that Bazooka overexpression does not influence its subcellular distribution dynamics. In contrast both the subcellular distribution and recruitment kinetics of BazΔ differed from the full length Bazooka. A vesicular, cytosolic pool of BazΔ was present in addition to the membrane

pool, and BazΔ recruitment to en-face DME interfaces was both delayed and less tight (11.8 ± 1.2 mins and 20.4 ± 2.2 mins for UAS Baz GFP and UAS BazΔ respectively; *Figure 6B*, *Figure 6—figure supplement 1F* and *Figure 6—video 2*). While en-face contacts quickly interlocked in genomic Bazooka or full-length Bazooka overexpressing stripes (15.9 ± 1.2, 16.4 ± 1.0 mins respectively for genomic Baz GFP, UAS Baz GFP; n = 41 and 47 interfaces respectively), expression of BazΔ pro-longed the time taken to interlock (24.6 ± 1.5 mins, n = 26 interfaces; *Figure 6—figure supplement 1E*). Whereas interlocked interface morphologies were evident 15 minutes after fusion in stripes from control embryos expressing sqh::Utrophin GFP, such morphologies could not be easily dis-cerned in the Baz RNAi expressing stripes even 45 minutes after fusion. Additionally, the fusing fronts of Baz RNAi expressing stripes remained enriched with actin (*Figure 6C–D* and *Figure 6—vid-eos 3A*, *4*). These results reveal that spatiotemporally regulated changes in Bazooka distribution modulate interface remodelling.

## Bazooka influences cell dynamics and epithelial organization in fusing stripes

Bazooka perturbations also influenced epithelial organization within the stripe. In Baz RNAi and BazΔ expressing stripes, cells in the stripe appeared to transgress the dorsal midline leading to an obliterated seam. In contrast, Bazooka overexpression within the stripe resulted in greater 'order', evident in the arrangement of cell rows (*Figure 5E* and *Figure 6—video 1*). These results suggest that Bazooka levels/function influence cell dynamics within the stripe. Cell shape (aspect ratio of the DME cells measured as ratio of D/V by A/P lengths) was significantly altered (p=0.018) upon Bazooka overexpression (mean aspect ratio 4.3 ± 1.3, n = 59 cells from six embryos) compared to control cells (mean aspect ratio 4.9 ± 1.3, n = 58 cells from seven embryos) but no significant differ-ences were observed in BazΔ or Baz RNAi expressing cells (BazΔ: 4.9 ± 1.5, n = 58 cells from six embryos and Baz RNAi: 4.6 ± 1.5 n = 42 cells from six embryos). However, both perturbations mar-ginally increased the number of cells with higher aspect ratio (≥6).

## Bazooka entrains actomyosin contractility during epithelial fusion

Two mechanisms have previously been suggested to explain the effect of Bazooka on cell behaviour and junction remodelling: i) the regulation of cell adhesion, and ii) the regulation of actomyosin con-tractility. We first examined the effects of Bazooka perturbations on ECadherin levels and distribu-tion (*Figure 5D–G*). Overexpression of Baz or BazΔ in stripes resulted in a reduction in ECadherin intensity in the entire stripe, giving it a distinct paler appearance, compared to either control or the unperturbed interstripes. In contrast, stripes expressing Baz RNAi did not show an obvious reduction in ECadherin intensity. While both control and Bazooka overexpressing DME cells maintained a mar-ginal anisotropic enrichment of ECadherin at the fused interfaces (compared to the D/V interfaces), ECadherin distribution in BazΔ and Baz RNAi expressing DME cells was more isotropic. Overall, the effect of Bazooka perturbations on ECadherin levels was modest at best.

We then examined the distribution of actin in Baz RNAi expressing stripes using sqh::Utrophin GFP. In contrast to control embryos, in which Utrophin GFP appeared to be significantly reduced at fused DME cell interfaces approximately 10 minutes after fusion (10.9 ± 0.9 mins, n = 5 embryos), the fusing fronts in Baz RNAi expressing cells were enriched with Utrophin GFP even 20 minutes (n = 5 embryos) after fusion (*Figure 6C–D* and *Figure 6—videos 3A*, *4*). This observation uncovers a role for Bazooka in actin remodelling or clearance during fusion and suggests that Bazooka might influence interface remodelling through the regulation of actin dynamics. The presence of GFP tags in the Bazooka overexpression transgenes did not allow the examination of Utrophin GFP at their DME cell interfaces.

## Bazooka dependent cell shape changes and rearrangements regulate fusion fidelity

Bazooka perturbations also influenced the modulation of fusing front width and cell number during the evolution to high fidelity. Notably, there was an increase in cell number disparity between part-ner stripes at fusion, with an increase in the proportion of stripes exhibiting a disparity of two cells with all three perturbations (23.5% and 41.6%, for UAS Baz and UAS BazΔ; and 0% for control; 0% for UTRN GFP and 16.6% for Baz RNAi; *Figures 6E–F* and *7D*). Bazooka perturbations also altered

the constriction index of the fusing front (width/cell number) with Bazooka overexpression producing a significant increase and BazΔ and Baz RNAi modestly reducing it (*Figure 6G*). Despite these changes, stripes overexpressing UAS Baz aligned by transient but pronounced changes in stripe width (*Figure 6A* and *Figure 6—video 1*). In contrast, stripes expressing BazΔ or Baz RNAi were kinked, exhibited overhangs at fusion and often contained more than one engrailed negative cell at the leading edge (*Figures 6B* and *7C,E* and *Figure 6—videos 2, 4*). In a small fraction of Baz RNAi expressing embryos, obvious mismatches of segments and epithelial tears were observed (*Figure 7B,E* and *Figure 5—figure supplement 1H–I*). These results strongly suggest that spatio-temporally regulated changes in Bazooka levels, distribution and/or function regulate the adaptive changes in cell number, cell rearrangements and cell shape to contribute to fusion fidelity, epithelial continuity and epithelial order.

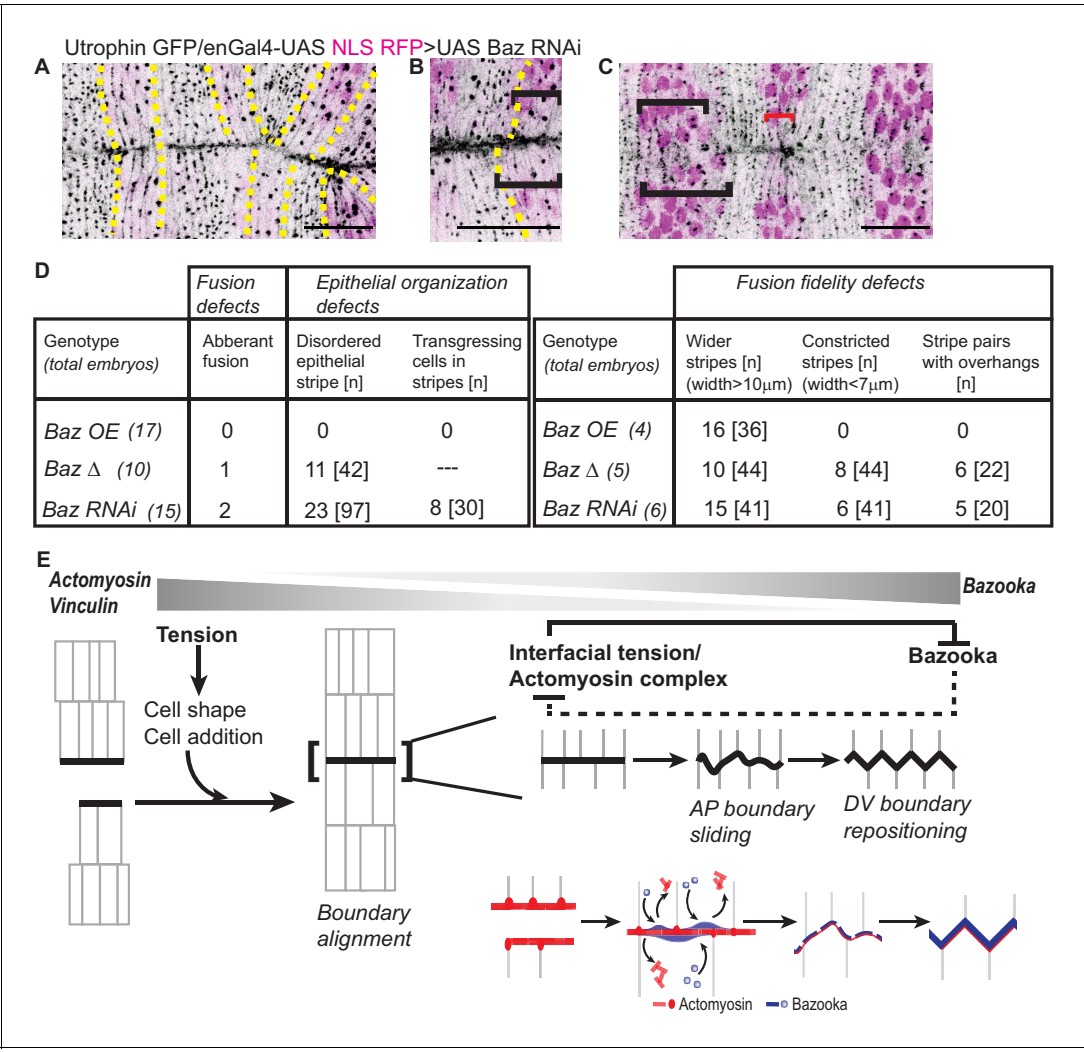

**Figure 7.** Bazooka downregulation reduces fusion fidelity and epithelial order. (**A–C**): Post fusion images of the dorsal midline from movies of embryos expressing UAS Baz RNAi driven by enGal4 (embryos also express Utrophin GFP and UAS NLS RFP) showing abnormalities in fusion fidelity including aberrant fusion (A, yellow dashed lines indicate the stripes), overhangs between contralateral partners at fusion (B, dashed yellow line marks the anterior boundary of the stripe) and stripe width disparities (C, black brackets indicate wider stripes and red bracket indicates constricted stripes). Scale bar- 20 μm. (**D**) Prevalence of defects in fusion, fusion fidelity and epithelial organization observed in the Bazooka perturbations tested. (**E**) Graphic summary of the cellular, subcellular, molecular and physical changes that enable interface remodelling and ensure fusion fidelity and epithelial continuity.

DOI: https://doi.org/10.7554/eLife.41091.027

## Tension-dependent Bazooka recruitment promotes intercalatory dynamics and interface remodelling in induced discontinuities

The work described above uncovers a role for the timely recruitment of Bazooka at the fusing interfaces for the molecular and cellular changes that confer fidelity to epithelial fusion. Earlier work on Drosophila germband extension uncovered the mutual exclusion of Bazooka and myosin from A/P and D/V interfaces of cells respectively and suggested mutual negative regulation (*Blankenship et al., 2006*; *Simões et al., 2010*). To determine whether cytoskeletal tension might influence the recruitment of Bazooka to DME interfaces, we used laser ablation to create a linear discontinuity within the stripe along its A/P axis during (pre-fusion) and after (post-fusion) the completion of dorsal closure in embryos overexpressing Bazooka in the stripe. This creates two flanks on the same side of the embryo. In pre-fusion stripes, such an ablation did not impede the fusion of the contralateral partners at the dorsal midline. The dorsal part of the cut stripe surged ahead after ablation, and remodelled its fusing interfaces with orderly Bazooka recruitment and the appearance of interlocked interfaces. The cut ends of the stripe were depleted of Bazooka for about 20 minutes after ablation when punctae of Bazooka became detectable. Notably, Bazooka recruitment was detected as the distance between the cut ends reduced, and preceded actual contact between the fusing interfaces (*Figure 6—figure supplement 1A–B* and *Figure 6—video 5*). These results suggest that tension release promotes Bazooka recruitment. Similar results were observed when stripes were cut post-fusion except that both the recruitment of Bazooka and the reestablishment of continuity were more rapid presumably due to the lack of interference from fusion as part of dorsal closure (*Figure 6—figure supplement 1C–D* and *Figure 6—video 5*). Remarkably in both cases, directed, intercalatory movements of cells within the cut stripes contributed to the establishment of a linear seam. The onset of muscle contraction however precluded the assessment of the precise nature of interface remodelling. These results suggest that Bazooka recruitment, which is influenced by tension, may be a generic regulator of epithelial continuity.

## Discussion

Much work has informed our understanding about the nature and origin of forces that drive cell sheet movements during fusion morphogenesis. Biophysical studies in the genetically and cell biologically tractable model, dorsal closure, have uncovered the remarkable resilience of the process to perturbations in the participating tissues and suggested that multiple forces contribute to its progression (*Hutson et al., 2003*; *Kiehart, 1999*; *Kiehart et al., 2000*; *Martin and Parkhurst, 2004*; *Martin and Wood, 2002*). While the cellular and molecular origins of the forces have been identified, how fidelity in fusion is achieved and how stable epithelial continuity is established is poorly understood (*Kiehart et al., 2017*). These are questions of fundamental importance. They address not only how contacts formed during re-epithelialisation become strong and stable but also how the physical sizes of (geometric/morphological symmetry and scaling) and the genetic pre-patterns in the fusing flanks are matched and maintained in the face of large-scale cell movements. In addition to establishing epithelial continuity and integrity, fusion morphogenesis also ensures the positions of cells of specific fates. A failure in fusion can thus have consequences not only on tissue integrity and morphology but also on tissue function. The work we describe here uncovers the cellular and molecular mechanisms that ensure fusion fidelity and impart mechanical integrity to the newly formed seam.

### Fidelity in epithelial fusion: the logic and the mechanisms

Our work uncovers for the first time, the principles that govern fidelity in epithelial fusion during *Drosophila* dorsal closure. Our findings demonstrate that fidelity does not rely on pre-set, invariant and equal parameters (DME cell numbers and fusing front geometry) in the two fusing flanks. Rather, spatially constrained adaptive mechanisms that primarily modulate the length and geometry of the fusing front result in the temporal evolution to high fidelity (*Figure 1—figure supplement 1Q*). Genetically hardwired mechanisms (segmentation genes) spatially constrain adaptive mechanisms (cell shape changes and cellular rearrangements) to distinct embryonic segments and segment compartments but do not predetermine the exact numbers of cells in each fusing segment or arc. The adaptive mechanisms are modulated by changes in cytoskeletal tension and are mediated by

anisotropic cell shape changes that alter fusing interface lengths, and possibly also by cell rearrangements. These mechanisms ensure that fused segments are matched in width despite a cell number disparity of one or two. Our findings thus argue that cell number matching during dorsal closure (*Jacinto et al., 2000*; *Millard and Martin, 2008*) operates at less than perfect fidelity.

## Establishing stable epithelial continuity by interface remodelling after fusion

Our results also uncover for the first time, changes in interface geometry that accompany fusion. These changes, characterized by the rapid remodelling of horizontal en-face interface configurations formed at fusion to angled interfaces that reposition along the D/V interfaces, result in the formation of tricellular vertices and cell interlocking across the fusing front. This conversion obliterates the straight seam evident at fusion, and we speculate, achieves a more stable junctional configuration that imparts mechanical integrity to the epithelial sheet (*Figure 7F*). This remodelling presumably also obviates the need to precisely match cell numbers at fusion.

## System variables, spatial constraints and the temporal evolution to high fidelity

Our quantitative spatiotemporal analysis of disparities in cell number and geometry in the fusing fronts revealed that neither parameter was pre-set to be identical in the two fusing fronts at the outset. Rather, significant reductions in disparity were observed over the course of closure, first in arc length and then in cell number, suggesting that the system evolves to become better matched. Our findings also uncover error correction mechanisms that primarily modulate of the width of the fusing front to ensure segment alignment even in the face of disparities in cell number that persist at fusion.

The observation that fusing front geometry and cell numbers are differentially regulated along the anterior-posterior axis of the embryo and within each embryonic segment suggests genetic control by homeotic and segment polarity genes. Indeed, homeotic genes of the Abd A/B complex have been shown to regulate cell mixing between anterior and posterior compartments in segments T3-A6, the same segments that contribute to the cell number changes we observe during the evolution of fidelity (*Gettings et al., 2010*; *Roumengous et al., 2017*). Thus, genetically hardwired mechanisms must spatiotemporally constrain cellular mechanisms that contribute to fidelity. Whether and how segment polarity genes confer differences in cell behaviour that explain the differential contributions of anterior and posterior compartments of embryonic segments to fusion fidelity remain to be determined.

Our results also reveal that cell rearrangements occur well before fusion, even in the absence of cell number disparity, and coincide temporally with width equalization. Conversely, disparities in cell number at fusion are corrected by rapid modulation of fusion front length. These results argue that both cell rearrangements and cell shape changes enable the equalization of geometry albeit at different times and with different efficiencies/speeds. The early cell number and width modulation we observe suggests that sensing mechanisms must operate at a distance within each segment compartment. The width adjustments that occur after fusion operate up close and align compartment boundaries. These findings suggest that distinct mechanisms separated in time and distinguished by different length scales of operation must contribute to sensing cell number and width within each compartment and enable D/V interfaces at the compartment edges to sense and find each other.

## Cytoskeletal tension as a trigger for adaptive mechanisms

Our work suggests that tension generated by the cytoskeleton may serve as a sensing mechanism that sets in motion adaptive mechanisms that contribute to fusion fidelity. Indeed, tension entrained the temporal evolution of all the fidelity parameters examined. Perturbations that lowered tension also affected segment matching in a small fraction of embryos examined, as had been previously observed in heterozygous *zip* (non muscle myosin heavy chain) mutants that also expressed ZipDN in the engrailed compartment (*Franke et al., 2005*). How and where tension is sensed and how it triggers these adaptive mechanisms are areas of future investigation.

## Bazooka: a key regulator of interface geometry

Our work uncovers the importance of modulating interface geometry for ensuring fusion fidelity and epithelial continuity. Interface length was modulated by anisotropic cell shape changes and ensured fusion fidelity even in the face of large cell number disparities induced by Bazooka overexpression. These results suggest a role for Bazooka in modulating junction length. Consistent with this, Bazooka colocalised with ECadherin at fusing interfaces (*Figure 5—figure supplement 1G*) and high intensity Bazooka GFP patches shifted dynamically along the length of the interface during junction remodelling, sparing the vertices or tricellular junctions (*Figures 2D–E* and *6A*, *Figure 2—figure supplement 1A*, *Figure 6—figure supplement 1A,C*, *Figure 2—video 4*, *Figure 6—video 1* and *Figure 6—video 5*). These findings suggest that the dynamic repositioning of Bazooka patches modulates interface length and geometry.

Interface remodelling accomplished the repositioning of A/P oriented en-face contacts and enabled cell interlocking. Our work demonstrates that this remodelling depends on the reciprocal spatiotemporal regulation of cytoskeletal tension and Bazooka. Bazooka has been previously shown to contribute to anisotropic cell shape change in the epidermis and to the resolution of unstable junctional intermediates during cell intercalation (*Bertet et al., 2004*; *Blankenship et al., 2006*; *Bulgakova et al., 2013*; *Simões et al., 2010*). The former has been shown to rely on the regulation of ECadherin turnover. The latter may rely on its ability to exclude myosin from resolving interfaces. Our results identify one potential mechanism that may contribute to Bazooka's effects on interface remodelling: the regulation of interfacial tension. Our results revealed that Bazooka and actomyosin contractility/cytoskeletal tension (actin, myosin and vinculin) were reciprocally regulated in space and time at the fusing/fused interfaces (*Figure 2H*). Notably, Utrophin GFP appeared to accumulate at fusing interfaces in cells lacking Bazooka (*Figure 6D*). These results tempt the speculation that Bazooka may mediate its effects on junction remodelling through the regulation of actomyosin contractility and through it, interfacial tension (*Figure 7E*). Additionally, Bazooka heterogeneities within a fusing interface may enable the modulation of tension along each interface. Thus, Bazooka may regulate the modulation of interface tension locally and its relative depletion at the vertices may help maintain high tension there. Indeed, Bazooka has been shown to interact either genetically or physically with myosin, the Rho1 GTPase effector ROCK and the *Drosophila* β-Catenin homolog Armadillo, and the effects we see on actin may be mediated by any of these interactions (*Simões et al., 2010*; *St Johnston and Sanson, 2011*). Our results also revealed that Bazooka downregulation has only a modest effect on ECadherin anisotropy. We have however not examined it effects on ECadherin turnover at the fusing DME cell interfaces, which is technically challenging on account of the fact that the interfaces are dynamic. What molecular interactions drive the effects of Bazooka we observe is a matter currently under investigation.

Bazooka also influenced epithelial organization, producing noticeable differences in cell packing within the stripe. While Bazooka over expression produced neatly ordered cell rows, both BazΔ and Baz RNAi increased cell dynamics. These results suggest that the relative distribution of Bazooka at cellular interfaces, which is both genetically hardwired (A/P vs D/V boundary differences) and emergent (fusing interface recruitment and dynamics within the interfaces), controls cell shape and cell dynamics through its effects on interface morphology. Whereas overexpressed Bazooka maintained these differential distributions, the recruitment of BazΔ specifically to the fusing DME interfaces was delayed, enabling us to examine the effect of dysfunctional Bazooka recruitment. The efficacy of BazΔ in disrupting fusion fidelity, epithelial organization and interface remodelling as well as its phenotypic similarities with Baz RNAi allow us to conclude that the spatiotemporal regulation of Bazooka entrains the temporal evolution of changes in interface length and geometry that contribute to fusion fidelity, seamless epidermal continuity and epithelial order.

## Adaptive mechanisms in the regulation of fusion fidelity

Initially identified as a regulator of apicobasal polarity, Bazooka's roles have expanded to include adhesion junction remodelling (during cell intercalation), anisotropic cell shape change and boundary positioning (during the formation of segment boundaries) (*St Johnston and Sanson, 2011*). Bazooka and myosin also exhibit reciprocally planar polarized patterns within the same cells and can influence each other negatively (*Blankenship et al., 2006*; *Simões et al., 2010*). Our ablation experiments suggest that release of interfacial tension is sufficient to recruit overexpressed Bazooka to the wound

edge interfaces and to drive healing by intercalatory movements and cell interlocking rather than by wound edge contraction. Combined with the effects that Bazooka knockdown produces on actin, our findings tempt the suggestion that a biomechanical feedback loop operating at the scale of single interfaces facilitates junctional changes that accompany fusion. In this feedback loop, Bazooka negatively influences actomyosin contractility in the DME cells, which in turn promotes Bazooka recruitment (*Figure 7F*). Potentially, such a mechanism can enable tension-dependent regulation of interface geometry, analogous to what has been suggested for the regulation of pulsed constrictions (*Munjal et al., 2015*). Together, out results identify Bazooka as a tension sensitive regulator of interface geometry. What molecular interactions promote Bazooka localization and dynamics at fusing interfaces in a tension sensitive manner and enable the modulation of interface geometry remain questions for future investigation.

The spatiotemporal regulation of tension also governs the directional temporal progression of fusion during neural tube closure in the ascidian *C. intestinalis.* The sequential fusion of individual cell pairs provides the force dynamics (relaxation) necessary for the fusion of the subsequent cell pairs (*Hashimoto et al., 2015*). Thus, junction relaxation may be a conserved feature in the progression of epithelial fusion. Junction relaxation and elongation have recently been demonstrated to be active processes that are regulated by directed cytoskeletal tension in neighbouring cells (*Bardet et al., 2013*; *Collinet et al., 2015*; *Hara et al., 2016*). Whether junction remodelling during fusion requires tension in neighbouring cells and whether Bazooka is a generic regulator of tension will be interesting to determine.

## Conclusions, Significance and Implications

Our work sheds light on the mechanisms that ensure epithelial continuity and fusion fidelity during epithelial fusion. Our results highlight the role of adaptive changes in cell behaviour - anisotropic cell shape changes and cellular rearrangements- in the refinement of spatial patterns established by gene regulatory networks to ensure morphological and molecular symmetry across the newly formed seam. It uncovers changes in interface geometry that contribute to the formation of a strong and stable seam. Our results also identify cytoskeletal tension and the polarity protein Bazooka as key regulators whose mutual dependence and exclusion might enable a self-organised biomechanical feedback mechanism that operates within an interface to entrain the dynamics of cells and cellular interfaces and ensure fusion fidelity and stable epithelial continuity. How general the cellular and subcellular mechanisms that we have identified are in other contexts of reepithelialisation will be interesting to determine.

# Materials and methods

**Key resources table**

| Reagent type (species) or resource | Designation | Source or reference | Identifiers | Additional information |
|---|---|---|---|---|
| Genetic reagent | | | | |
| *D. melanogaster*: ubi::ECadherin GFP | Drosophila stock | *Oda and Tsukita, 2001*. Kind gift of Tadashi Uemura, Kyoto University, Japan | Flybase_FBtp0014096 | |
| *D. melanogaster*: Bazooka GFP | Drosophila stock | *Buszczak et al., 2007* Bloomington Drosophila Stock Centre (BDSC) | BDSC_51572 FlybaseFBst0051572 | |
| *D. melanogaster*: y w sqh^AX3; sqh GFP | Drosophila stock | Bloomington Drosophila Stock Centre (BDSC) | BDSC_57144 Flybase_FBst0057144 | |
| *D. melanogaster*: sqh::Utrophin GFP | Drosophila stock | *Rauzi et al., 2010*. Made by Thomas Lecuit, IBDM, Marseille, France, obtained from Richa Rikhy, IISER Pune, India | Flybase_FBal0270167 | |

*Continued on next page*

*Continued*

| Reagent type (species) or resource | Designation | Source or reference | Identifiers | Additional information |
|---|---|---|---|---|
| *D. melanogaster*: UAS ZipperDN YFP | Drosophila stock | *Dawes-Hoang et al., 2005*. Kind gift of Andrea Brand, University of Cambridge, UK | Flybase_FBtp0021364 | |
| *D. melanogaster*: UAS SqhAA | Drosophila stock | *Jordan and Karess, 1997*. Bloomington Drosophila Stock Centre (BDSC) | BDSC_64114 Flybase_FBst0064114 | |
| *D. melanogaster*: UAS Bazooka GFP | Drosophila stock | *Benton and St Johnston, 2003*. Kind gift of Daniel StJohnston, University of Cambridge University, UK | Flybase_FBtp0017662 | |
| *D. melanogaster*: UAS Bazooka GFPΔ969–1464 | Drosophila stock | *Krahn et al., 2010*. Kind gift of Richa Rikhy, IISER Pune, India | Flybase_FBtp005 850 | |
| *D. melanogaster*: UAS Actin5C GFP | Drosophila stock | Bloomington Drosophila Stock Centre (BDSC) | BDSC_9257 Flybase_FBst0009257 | |
| *D. melanogaster*: UAS Bazooka RNAi | Drosophila stock | *Weng and Wieschaus, 2017*. Bloomington Drosophila Stock Centre (BDSC) | BDSC_35002 Flybase_FBst0035002 | |
| *D. melanogaster*: engrailedGal4 | Drosophila stock | Bloomington Drosophila Stock Centre (BDSC) | BDSC_30564 Flybase_FBst0030564 | |
| *D. melanogaster*: $w^{1118}$ | Drosophila stock | Bloomington Drosophila Stock Centre (BDSC) | BDSC_5905 Flybase_FBst0005905 | |
| *D. melanogaster*: enGal4-ubi:: ECadherin GFP | Drosophila stock | Recombinant, this study | N/A | |
| *D. melanogaster*: enGal4-UAS GFP | Drosophila stock | Recombinant, this study | N/A | |
| *D. melanogaster*: enGal4-UAS NLS RFP | Drosophila stock | Bloomington Drosophila Stock Centre (BDSC) | BDSC_30557 Flybase_FBst0030557 | |
| *D. melanogaster*: Vinculin GFP | Drosophila stock | Vienna Drosophila Resource Centre (VDRC) | v318227 Flybase_FBst0491774 | |
| *D. melanogaster*: sqh::Utrophin GFP; UAS Baz RNAi | Drosophila stock | This study | | |
| *D. melanogaster: baz⁴* | Drosophila stock | Bloomington Drosophila Stock Centre (BDSC) | BDSC_23229 Flybase_FBst0023229 | |
| *D. melanogaster*: UAS SqhEE | Drosophila stock | Bloomington Drosophila Stock Centre (BDSC) | BDSC_64411 Flybase_FBst0064411 | |
| *D. melanogaster*: LEGal4 | Drosophila stock | Bloomington Drosophila Stock Centre (BDSC) | BDSC_58801 Flybase_FBst0058801 | |
| Antibodies | | | | |
| Anti GFP A11122 | Primary | Invitrogen | AB_221569 | |
| DCAD2 (Anti ECadherin) | Primary | DSHB | AB_528120 | |
| Anti Vinculin N19 (SC7649) | Primary | Santa Cruz Biosciences | AB_2288413 | |

## Drosophila stocks

ubi::ECadherin GFP [ECadh GFP; *Oda and Tsukita, 2001*], Bazooka GFP [Baz GFP, Bl 51572; *Buszczak et al., 2007*], *y w sqh*$^{AX3}$; sqh GFP (sqh GFP, Bl 57144) (all from Bloomington Drosophila Stock Centre, Indiana University, USA) and sqh::Utrophin GFP [Utrophin GFP/UTRN GFP; (*Rauzi et al., 2010*); a kind gift from Richa Rikhy, IISER Pune] were used to visualize junction and cytoskeletal dynamics during fusion. UAS ZipperDN YFP [UAS ZipDN; kind gift from Andrea Brand, University of Cambridge; (*Dawes-Hoang et al., 2005*), UAS SqhAA (Bl 64114) and UAS SqhEE (Bl 64411) [(*Jordan and Karess, 1997*), both from Bloomington Drosophila Stock Centre],

UAS Bazooka GFP [UAS Baz GFP/UAS Baz; kind gift from Daniel St Johnston, University of Cambridge; (*Benton and St Johnston, 2003*)], UAS Bazooka RNAi [UAS Baz RNAi, Bl 35002; (*Weng and Wieschaus, 2017*) (from Bloomington Drosophila Stock Centre) and UAS Bazooka GFP Δ969–1464 [UAS BazΔ; kind gift from Richa Rikhy, IISER Pune; (*Krahn et al., 2010*)] were driven with engrailed Gal4 (enGal4, Bl 30564) or enGal4-UAS NLS RFP (Bl 30557)(both from Bloomington Drosophila Stock Centre) to perturb cytoskeletal tension or Bazooka levels in the posterior compartment of embryonic segments (stripes) along with UAS GFP or UAS Actin5C GFP (UAS Actin GFP, Bl 9257, Bloomington Drosophila Stock Centre). $baz^4$ (Bl 23229, Bloomington Drosophila Stock Centre) was also used to examine the effect of loss of zygotic Bazooka in embryos. Stable enGal4-ubi::ECadherin GFP and enGal4-UAS GFP recombinants were built to visualize the epidermal cells and the stripes. LEGal4 (Bl 58801, from Bloomington Drosophila Stock Centre) was used to drive SqhAA, SqhEE or ZipDN in the DME cells. The stock sqh::Utrophin GFP/CyO; UAS Baz RNAi/TM3SerGFP was built to visualize actin organization upon downregulating Bazooka. Vinculin GFP (VDRC v318227) was used to determine the distribution of Vinculin.

## Genotypes examined

*Figure 1*, *Figure 1—figure supplements 1*, *2* and *3*: *w*; enGal4-ubi::ECadherin GFP/+; UAS Actin5C GFP/+ to estimate epidermal arc lengths, stripe width and DME cell number.

*Figure 2A w*; ubi::ECadherin GFP was used to visualize ECadherin.

*Figure 2B w*; sqh::Utrophin GFP/+ was used to visualize actin localization in stripes and interstripes.

*Figure 2C y w sqh^{AX3}*; sqh GFP was used to visualize myosin.

*Figure 2D* and *Figure 2—figure supplement 1A*: $y^1$ *w* Baz GFP was used to examine native Bazooka localization.

*Figure 2E* $y^1$ *w* Baz GFP/+; enGal4-UAS NLS RFP/+ was used to visualize Bazooka localization in stripes and interstripe.

*Figure 2F w^{1118}* was used to examine Vinculin localization.

*Figures 3* and *4A–C* and *Figure 3—figure supplement 1*: *w*; enGal4-ubi::ECadherin GFP/+; UAS ZipDN/+ was used to perturb tissue tension. *w*; enGal4-ubi::ECadherin GFP/+; UAS Actin5C GFP/+ was used as a control.

*Figures 4D–E* and *5A–C*: *w*; enGal4/+; UAS ZipDN/+ and *w*; enGal4-UAS GFP/+; UAS SqhAA/+ were used to perturb tissue tension. *w*; enGal4-UAS GFP/+ was used as a control.

*Figure 5D–G w*; enGal4/UAS Baz GFP and *w*; enGal4/+; UAS Baz GFPΔ969–1464/+ and *w*; enGal4-UAS GFP/UAS Baz RNAi were used to perturb Bazooka levels/function. *w*; enGal4-UAS GFP/+ was used as a control.

*Figure 6A* and *Figure 6—figure supplement 1A*: *w*; enGal4/UAS Baz GFP was used to visualize Bazooka localization and examine the effect of overexpression in real time with or without ablation.

*Figure 6B w*; enGal4/+; UAS Baz GFPΔ969–1464 /+ was used to perturb Bazooka function and examine its effect in real time.

*Figure 6C w*; sqh::Utrophin GFP/enGal4-UAS NLS RFP was used to visualize stripe cell shape and interface morphologies and actin organization at the leading edge in real time in an unperturbed condition.

*Figures 6D* and *7A–D*: *w*; sqh::Utrophin GFP/enGal4-UAS NLS RFP; UAS Baz RNAi/+ was used to down-regulate Bazooka function in stripes and examine its effect upon interface remodelling and actin organization in real time.

*Figure 2—figure supplement 1B w*; sqh::Utrophin GFP/enGal4-UAS NLS GFP used to visualize actin dynamics in stripes and interstripes.

*Figure 2—figure supplement 1C,D* Vinculin GFP was used to determine the distribution of Vinculin during fusion.

*Figure 5—figure supplement 1A–D w*; LEGal4/+; UAS ZipDN/+, *w*; LEGal4/+; UAS SqhAA/+ and *w*; LEGal4/+; UAS SqhEE/+ were used to perturb cytoskeletal tension in the DME cells. *w*; LEGal4/UAS GFP was used as a control.

*Figure 5—figure supplement 1E–F w*; enGal4-UAS GFP/+; UAS SqhAA/+ was used to assess the distribution of interface morphologies and angles. *w*; enGal4-UAS GFP/+ was used a control.

*Figure 5—figure supplement 1G w*; enGal4/UAS Baz GFP was used to determine Bazooka localization in the DME cells.

*Figure 5—figure supplement 1H–I w*; enGal4-UAS GFP/UAS Baz RNAi was used to knock down Bazooka in stripes.

*Figure 5—figure supplement 1J y$^1$ w baz$^4$* was used to assess the effect of zygotic Bazooka loss of function on junction geometries.

*Figure 6—figure supplement 1E–F* w; enGal4/UAS Baz GFP and w; enGal4/+; UAS Baz GFPΔ969–1464/+ was used to assess the time taken to interlock and the time of appearance of Bazooka at fused interfaces in the Bazooka perturbations. *y$^1$ w* Baz GFP was used as a control in *Figure 6—figure supplement 1E*.

## Staging of embryos

Embryos were harvested from a 12–16 hr collection. For live imaging, embryos in which the spiracles were completely visible on the dorsal surface and the distance between the contralateral stripes of segments A8 and head segment Lb was approximately 30 µm apart were chosen. The scalloped leading edge of the epidermis was just visible in a dorsal-up view. Also, the anterior end of the hindgut was positioned beneath the amnioserosa at approximately the posterior 1/3$^{rd}$ of the distance between the two canthi. Embryos in which all embryonic segments had fused, the hindgut was completely uncoiled, and the yolk showed three lobes were used as post-closure stage embryos in fixed preparations (*Figure 5* and *Figure 5—figure supplement 1*).

## Immunohistochemistry

Embryos were fixed essentially as described (*Narasimha and Brown, 2006*) and mounted using Vectashield mounting medium (Vector Laboratories, Inc, CA). The following primary antibodies were used: anti-ECadh antibody (DCAD2, 1:10; Drosophila Studies Hybridoma Bank), rabbit anti-GFP antibody (A11122, 1:1000; Invitrogen) and anti-Vinculin antibody (SC 7649, 1:50; Santa Cruz Biosciences). The following secondary antibodies were used: Alexa Fluor anti-rat 633 (A21094), anti-rabbit 488 (A11034) and anti-goat 568 (A11057) (1:200, Invitrogen). The stained embryos were imaged using 60X oil (NA 1.4) immersion lens on an Olympus FV 1200 confocal microscope. Approximately 20–25 optical slices, 0.3 µm apart were used for maximum intensity projections on ImageJ. Figures were prepared using Adobe Photoshop and Adobe Illustrator (Adobe Systems, USA).

## Live imaging

The selected embryos were placed on 22 mm X 40 mm cover glass (Corning Technologies) in a thin film of Halocarbon oil 700 (Sigma Life Sciences) and imaged using either a Zeiss 710 Meta confocal microscope (Carl Zeiss, Germany) and a Plan-Neufluar 63X oil immersion objective (NA 1.4) or an Olympus FV1000 (60X oil immersion objective, NA 1.4). Optical sections 1–1.2 µm apart that captured the entire amnioserosa, leading edge and DME cells were acquired at a temporal resolution of approximately one to four 3D frames/minute for studies on fusing front length and dynamics. Maximum intensity projections (MIP) at each time point were made using the Image5D plugin of ImageJ and assembled as a time series using ImageJ. For the visualization of molecular dynamics, a smaller field of view encompassing 3–4 central fusing stripes were imaged with a 2–2.5X zoom and projections of 7–8 slices (step size 0.3–1 µm) were used to visualize the geometry of the fusing front. Figures were prepared using Adobe Photoshop and Adobe Illustrator (Adobe Systems, USA).

## Quantitative morphodynamic analysis

The parameters and formulae used to quantify the fidelity of fusion are shown in *Figure 1C*. The measurements were made as described below.

### Difference in arc and stripe length and cell number

The outlines of the arcs formed by the leading edge or fusing front (from the anterior canthus to the posterior canthus) of each epidermal flank were traced out manually using ImageJ from 190 minutes prior to fusion to the end of dorsal closure. The average of six measurements for every frame was used. The normalized arc length difference at a given time t was calculated as follows.

$$\text{Norm.}\Delta\text{Arclength} = (L_u - L_l)/\langle L \rangle \tag{1}$$

$L_u$ and $L_l$ are the lengths of the upper arc and the lower arc. $\langle L \rangle$ is the average of the two arc lengths at each time point and is determined by the formula:

$$<L> = (L_u + L_1)/2 \tag{2}$$

Similarly, the width at the leading edge or fusing front of each stripe or interstripe was traced out manually as the distance between the compartment boundaries identified by GFP, YFP or RFP. Measurements were made at every time frame starting from 100 minutes prior to fusion ($t_{-100}$) for the central stripes and from the beginning of dorsal closure (early time point/$t_E$) for the peripheral stripes to the end of fusion of the fusing pair ($t_F$). The two starting points were necessary since the peripheral stripes remained invisible in a dorsal-up view 100 minutes prior to their fusion. As a result, the time intervals over which the change in width is measured is different for the central and peripheral stripes (*Figure 1—figure supplement 1K and L*). For all other analyses, only the central stripes (A2-A4 for *Figure 1K* and A2-A4 for *Figure 3E–G*) were analysed and the differences in width in the 100 minute interval were calculated. The normalized width difference at a given time was calculated as follows.

$$\text{Norm.}\Delta\text{Width} = (W_u - W_1)/<W> \tag{3}$$

$W_u$ and $W_l$ are the widths of the stripe or interstripe from the upper flank and the lower flank respectively. $<W>$ is the average width of the contralateral partners at each time point and is determined by the formula:

$$<W> = (W_u + W_1)/2 \tag{4}$$

To determine the difference in total DME cell number between the two fusing flanks, the total number of DME cells in each epidermal flank was counted at five-minute intervals from 100 minutes prior to fusion to the end of dorsal closure. Whole cells that were located between the two canthi were counted and the average of six counts was used. DME cell number counts at earlier time points could not be obtained since the DME cells were not distinctly visible in a dorsal-up view. The normalized difference in DME cell number at a given time was calculated as follows.

$$\text{Norm.}\Delta\text{DME} = (N_u - N_1)/<N> \tag{5}$$

$N_u$ and $N_l$ are the total number of DME cells between the anterior and posterior canthi in the upper and the lower flanks respectively. $<N>$ is the average number of DME cells in the two arcs at each time point and is calculated as follows:

$$<N> = (N_u + N_1)/2 \tag{6}$$

DME cell numbers at the leading edge or fusing front of the stripe (the labelled GFP/RFP positive compartment) or interstripe (the unlabelled compartment between two GFP/RFP positive compartments) were counted at five-minute intervals starting from 100 minutes prior to fusion ($t_{-100}$) for the central stripes and from the early time point $t_E$ (beginning of dorsal closure) for the peripheral stripes (sometimes excluding the Lb/A7 segment on account of non-visibility) to the end of fusion ($t_F$) of the fusing pair (*Figure 1—figure supplement 1A–B and E–J*). For all other analyses, only the central stripes (A2-A4 in *Figures 1F–H* and *6E–F*) were analysed and the differences in cell number in the 100 min interval were calculated. For stripe DME cell numbers, all cells within the labelled compartment including any GFP negative cells contained within it were counted. The normalized difference in DME cell number of a pair of stripes/interstripes (from central segments A2, A3 and A4) at any given time t was calculated as follows.

$$\text{Norm.}\Delta\text{DME} = (n_u - n_1)/<n> \tag{7}$$

$n_u$ and $n_l$ are the DME cell numbers in the stripes or interstripes from the upper flank and lower flank. $<n>$ is the average number of DME cells in the fusing pair of stripes or interstripes at time t and is calculated as follows.

$$\langle n \rangle = (n_u + n_l)/2 \tag{8}$$

The image contrast was artificially increased post-acquisition for clear visualization of interstripe DME cells during counting (*Figure 1—figure supplement 1O–P*).

## Variance of ΔDME of stripe/interstripes

For each embryo, three pairs of contralateral partner and four pairs of ipsilateral partner stripes and interstripes belonging to segments A2, A3 and A4 were used. The variance of the difference in DME cell numbers (ΔDME) between the contralateral partners in an embryo was calculated as follows.

$$\sigma^2 = \frac{\sum(X - \mu)^2}{N} \tag{9}$$

$\sigma^2$ is the variance in ΔDME of partner pairs at a given time t. X is the ΔDME of each partner pair at time t. μ is the mean ΔDME and N is the sample size.

The average variance of ΔDME of partner pairs from multiple embryos was calculated as follows.

$$\mathrm{Var.\,\Delta DME} = \sum \sigma^2 / v \tag{10}$$

$\sigma^2$ is the variance in ΔDME in each embryo (*equation 9*) and ν is the total number of embryos.

## Alignment of stripes

A straight line joining the anterior ends of the posterior compartment of segment A3 between the upper and lower flank was drawn and the angle α subtended by it to the horizontal line (A/P axis) was measured using ImageJ. The degree of alignment (θ) was measured as follows.

$$\theta = I\,90° - \alpha\,I \tag{11}$$

## Constriction index of stripes

The width of the fusing fronts (W) as well as the number of DME cells (N) in one of the two flanks of the fusing stripe pair from the central segments (A2-A4) was measured at fusion. The constriction index for each stripe was calculated by the formula:

$$\mathrm{Constriction\,index} = W/N \tag{12}$$

## Intensity measurements

To measure ECadherin or actin intensity at the fusing interfaces, a line was drawn along the fusing A/P interfaces of ten (for ECadherin intensity) or five (for actin intensity) DME cells in the pre-fusion stage and mean intensity was measured using ImageJ. Their intensities at the midline seam were similarly determined at fusion (en-face contacts) and after fusion (interlocked interfaces). The mean fluorescence intensity (I) at each stage was calculated and normalized to the intensity of ECadherin or actin at the D/V oriented interfaces ($I_{DV}$) of the same cells.

$$\mathrm{Norm.\,intensity} = I/I_{DV}$$

## Estimation of types of cell rearrangements

Cell rearrangments and cell number changes at the leading edges of the stripes of segments T2 to A6 were visualised and determined by live imaging of enGal4-ubi::ECadherin GFP/+; UAS Actin5C GFP/+ embryos. Based on the ΔDME between partner stripes before and after the observed cell rearrangement, they were divided into the following types: (i) Type 1: ΔDME = 0 at all times, no addition in either partner; (ii) Type 2: ΔDME remains unchanged, addition occurs in both partner stripes; and (iii) Type 3: ΔDME reduces, addition occurs in the stripe with the lower cell number. To assess the change in stripe width upon cell addition, widths of contralateral partner stripes were measured at three instances: a) approximately 10 min prior to (pre-CR), b) immediately after (at CR) and c) 10 min after (post-CR) cell addition is complete in both stripe partners. The normalized disparity in width was then calculated as given in *equation 3* above.

## Assessing interface remodelling defects

To assess the defects in cell interlocking in tension defective fixed embryos (*Figure 5A–C*), the following stripes or interstripes were considered: (i) stripes (or interstripes) that were at least 10 cells away from the canthi in a closure stage embryo and (ii) all the stripes (or interstripes) except the central most stripe (or interstripe) in a post closure embryo. Stripes or interstripes from segments Lb and A7 were not considered. A stripe (or interstripe) was categorized as interlocking defective only

when at least 70% of the cells in that compartment (analysed on one side of the midline) lacked interlocked (Y-shaped/tricellular) interfaces.

In fixed embryos expressing UAS Baz GFP, UAS BazΔ or UAS Baz RNAi in the stripes (*Figure 5D–G*), the geometries of the fusing interfaces were classified into three categories: (i) 'angular' if the interfaces subtended an angle between 20° and 90° to the A/P axis; (ii) 'straight' if the interfaces subtended an angle between 0° and 20° to the A/P axis and (iii) 'curved' if the interfaces exhibited concave/convex curvatures or squiggles. Interfaces connecting two D/V interfaces and of at least 1 μm in length within stripes that belonged to segments T2-A6 were considered for analysis.

## Measurement of interface angles

A straight line was drawn along the interfaces (described above) between two vertices. The acute angle that this line subtended to the horizontal was measured using ImageJ.

## Laser ablation

Stripes of segment A2/A3 of embryos expressing Baz GFP in the stripe (enGal4/UAS Baz GFP) were chosen for laser ablation. Laser pulses (at 800 nm wavelength) were generated by a titanium-sapphire pulsed femtosecond laser (MaiTai DeepSee, Spectra Physics) coupled to a Zeiss 710 Meta microscope and delivered on to the sample using a 63X1.4 NA oil immersion objective essentially as described previously (*Meghana et al., 2011*). The laser power at the sample plane measured using a 10X 0.3 NA dry objective was approximately 720 mW. For ablations, 18% laser power was used with 18 iterations (pixel dwell time/iteration = 8.15μs) over a rectangular ROI positioned a row ventral to the DME cells. The ROI spanned the width of the stripe (approx. 10–15 μm) and was 1.05 μm in height. After ablation, the entire stripe was imaged for approximately 35 to 60 min at a temporal resolution of four 3D frames/minute. To assess the distance between the cut edges, a straight line was drawn between the furthest points on the two edges post ablation (approximately between the centers of the two cut edges) and the length of this line was measured at every time point using ImageJ.

## Statistical analysis

All statistical analyses were done using GraphPad Prism six software. Cluster plots show median ± interquartile range (the mean is indicated by the red dot in *Figure 3—figure supplement 1E–F*). The Mann-Whitney Unpaired t-test was used to assess the significance of difference between central tendencies. The f-test was performed to assess the significance in variance between the sets of data in *Figure 1G*. For *Figures 1*, *2A–B*, *3C–G*, *4C* and *6E–G*, *Figure 1—figure supplements 1* and *3E–F*, *Figure 3—figure supplement 1B–F* and *Figure 6—figure supplement 1E–F*, data from multiple live imaging sessions were pooled since it was not possible to analyze a statistically significant number of embryos from a single imaging session. Bar graphs show mean ± sd (*Figure 4C*, *Figure 1—figure supplement 1N*, *Figure 3—figure supplement 1C*) or mean ± sem (*Figure 5H–J*, *Figure 1—figure supplements 1A–B, K–L* and *3F* and *Figure 5—figure supplement 1E*). No statistical method was employed to predetermine the sample size. No data were excluded from the analysis.

## Acknowledgements

We thank Andrea Brand, Thomas Lecuit, Richa Rickhy, Daniel St Johnston, Tadashi Uemura, the Bloomington Drosophila Stock Center and the Developmental Studies Hybridoma Bank for reagents, FIJI/ImageJ (NIH) for image analysis software, Himanshu Sinha for suggestions for statistical analysis, Modhura Ganguly and Somnath Ghosal for providing some immunostained embryos, members of the MN lab and the Department of Biological Sciences for discussions, and Sudeepa Nandi and Anwesha Guru for comments on the manuscript. MN was supported by intramural funds and PTDG by a studentship from the Tata Institute of Fundamental Research/Department of Atomic Energy (DAE) of the Government of India.

## Additional information

### Funding

| Funder | Grant reference number | Author |
|---|---|---|
| Tata Institute of Fundamental Research | Studentship | Piyal Taru Das Gupta |
| Department of Atomic Energy, Government of India | Intramural Funds | Maithreyi Narasimha |

The funders had no role in study design, data collection and interpretation, or the decision to submit the work for publication.

### Author contributions

Piyal Taru Das Gupta, Resources, Formal analysis, Investigation, Visualization, Methodology, Writing—review and editing, Performed all experiments, Analysed data, Prepared figures, Contributed to manuscript editing and reviewing; Maithreyi Narasimha, Conceptualization, Resources, Formal analysis, Supervision, Funding acquisition, Investigation, Visualization, Methodology, Writing—original draft, Project administration, Writing—review and editing, Conceived, designed and supervised the project, Acquired funding, Contributed to the image acquisition and analysis, Wrote the manuscript

### Author ORCIDs

Maithreyi Narasimha (iD) http://orcid.org/0000-0001-8398-8023

### Decision letter and Author response

Decision letter https://doi.org/10.7554/eLife.41091.030
Author response https://doi.org/10.7554/eLife.41091.031

## Additional files

### Supplementary files

• Transparent reporting form
DOI: https://doi.org/10.7554/eLife.41091.028

### Data availability

All data generated or analysed during this study are included in the manuscript and supporting files. All measurements (individual data points) are presented in most of the graphs provided. Where a representative graph is shown in a figure, additional graphs are provided in figure supplements.

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
