## [Decision Letter]

[Editors’ note: a previous version of this study was rejected after peer review, but the authors submitted for reconsideration. The first decision letter after peer review is shown below.]

Thank you for submitting your work entitled "Cytoskeletal tension and Bazooka/Par3 modulate interface geometry to ensure fusion fidelity and seamless epithelial continuity during *Drosophila* dorsal closure" for consideration by *eLife*. Your article has been reviewed by three peer reviewers, one of whom is a member of our Board of Reviewing Editors, and the evaluation has been overseen by a Senior Editor. The reviewers have opted to remain anonymous.

Our decision has been reached after consultation between the reviewers. Based on these discussions and the individual reviews below, we regret to inform you that your work will not be considered further for publication in *eLife*.

All reviewers agree on the fact that there are interesting results presented in this manuscript, but that they are "buried", and the manuscript is difficult to read. Therefore, we must decline to consider the paper further in its present form. If the authors are able to address the points made by the referees below and write the manuscript in a way that allows an easy understanding, we would be willing to consider a revised version. This would be considered as a new submission with no guarantee of acceptance.

Reviewer #1:

Dorsal closure in the *Drosophila* embryo has been used extensively as a model to unravel the genetic, molecular and biophysical principles of epithelial morphogenesis. During this process, the two lateral epidermal sheets move dorsalwards and eventually fuse at the dorsal midline. Up to now, very little is known about the fusion process, in particular with respect to the precision by which cells of both sides recognise each other in order to form a uniform epithelial sheet.

In this manuscript, the authors set out to describe this process in great detail, focusing on the number and shape of the dorsalmost epidermal cells (DME) and the length covered by cells in stripes and interstripes, i. e. in the posterior and anterior compartments of the segments, respectively. Interestingly, they found that precision emerges over time, in particular with respect to the arc length, but that a certain degree of "imprecision" with respect to the cell number in contralateral stripes/interstripes is tolerated. Addition of cells from neighbouring compartments or from more ventral areas reduced the discrepancy in cell number in only some cases. They move on to show that the geometry of the junctions after fusion changes, from an "en-face" geometry to an "interlocked" pattern. This change is associated with a decrease of actin, Sqh and vinculin expression, being high in unfused cells, lower in en-face fused cells and even lower in "interlocked" cells. They further show that the proper change in junctional geometry depends on tension. As shown in other processes, e.g. germ band extension, expression of Bazooka (Baz)/Par3 shows an opposite behaviour, in that it is highest in "interlocked" cells. Upon overexpressing a truncated version of Baz, which is unable to localise at the membrane, this process is affected and goes along with reduced ECadherin expression, precocious interlocking, increased cell number disparity and changes in stripe width. Overall, most of the data presented here, which were obtained by an extremely careful analysis, add novel very important aspects for understanding epithelial fusion, showing that precision is not pre-set, but emerges.

The experiments are carefully described, and the data are very well presented. However, I have a problem to follow the conclusion made on the role of Baz in this process. From the experiments presented in Figure 5E-5G they suggested that "Bazooka enrichment is associated with reduced interfacial tension". To corroborate this assumption, they first showed that impaired tension (achieved by expression ZipDN or SqhAA) affects junction remodelling. Do the authors have any idea why dominant negative Zip and Sqh have different effects on the process?

They go on to express either GFP tagged full-length Baz, a GFP tagged C-terminally truncated Baz (Baz[Δ969-1464]) or Baz-RNAi. Expression of either full-length Baz or Baz[Δ969-1464] resulted in similar phenotypes, including weaker ECadherin staining, whereas expression of Baz-RNAi had only mild effects, but showing increased E-Cad staining. Baz-GFP was recruited with a similar dynamics as endogenously-tagged Baz. In contrast, Baz[Δ969-1464] recruitment was delayed, which was associated with an extended time required to reach the interlocked stage. Finally, they induced a wound in the epidermis on one side of the embryo and observed the fusion event in the wound. From all this they conclude that "… changes in interface length and geometry are dependent on regulation of cytoskeletal tension and Baz/Par3" (Abstract). While the data are convincing with respect to the importance of cytoskeletal tension in this process, the role of Baz is unclear. The fact that overexpression of Baz versions, in particular the truncated one, impairs this process, does not imply that Baz plays a role in this process, given that expression of RNAi has a different outcome. Rather, expressing this form results in a dominant phenotype, but does not provide convincing evidence on the role of Baz in this process.

Reviewer #2:

This manuscript by Das Gupta and Narasimha employs an elegant combination of genetics, live imaging, and morphometric analyses to study re-epithelialization in *Drosophila* dorsal closure. The authors first perform a detailed descriptive analysis of segment matching in terms of cell number and cell shape and uncover a role for myosin regulation (or, in their terms, tension) in this process. The authors then explore the resolution of junctional geometry following fusion. The results suggest that reciprocal regulation of Bazooka/Par-3 and vinculin localization downstream of cytoskeletal tension are required to effectively restore epithelial integrity.

While the study includes an impressive volume of detailed, quantitative analyses, the manuscript suffers from poor organization and lack of structure, which makes it challenging for the reader to tease apart the real significance of the authors' contributions. The finding that cell numbers between opposing sides of the embryo are not initially precisely regulated is interesting, as are the authors' data on reorganization of junctions following fusion. However, these two components of the story have not been well melded together, and it is difficult for the reader to see the forest for the trees. Despite these qualms, the videos are stunning, and the experiments are, for the most part, well controlled and carefully presented in the figures and captions-too much so at times, in terms of the amount of detail presented.

Major Concerns:

1) In the Introduction and Discussion section, the authors could do a better job explaining how the aspects of dorsal closure they are studying might have implications beyond this particular system. Concepts key to the authors' work, such as fidelity and precision, as they apply to this particular system, should be more clearly defined in the Introduction.

2) The authors perturb myosin as a way to make inferences about the role of tissue tension in this process. However, their strategy is somewhat problematic, as it relies on constitutive expression of a DN-zipper construct in engrailed stripes. While it is likely that tension does play a role in this process, it is difficult to interpret whether the observed changes in precision parameters occur because of a direct role for actomyosin-based tension in cellular rearrangements (which I believe that the authors want to imply) or that other aspects such as cell proliferation might be impaired. To address this problem, the authors should treat embryos with ROCK inhibitor Y-27632 during dorsal closure, which will allow more temporal control over manipulation of contractility.

3) As written, the manuscript is largely descriptive and difficult to digest. The novel points of the paper, although beautiful, are subtle and might be lost on readers unfamiliar with the extensive literature on dorsal closure. In its current form, this study may be better suited for a more specialist audience than the readers of *eLife*. This outcome would be a shame because the problem of re-epithelialization is of broad interest, and the authors' observations regarding interface remodelling after fusion are interesting, but they are regrettably buried in a sea of difficult-to-follow quantifications of unclear broader biological significance.

4) To begin to unclutter the story, Figure 1 and Figure 2, a tour de force of measurements and quantifications, could be simplified, with many of the detailed comparisons between stripes and interstripes removed or moved to supplemental. Similarly, Figure 7 and Figure 2—figure supplement 1 could be presented with fewer time points, just those illustrating the authors points. It is hard for the reader to digest so many panels.

Reviewer #3:

This manuscript focuses on the important problem of fusion fidelity and seamless epithelial continuity in the dorsal closure model for epithelial cell sheet migration and fusion. The authors find that differences in the length of the leading edge (here referred to as "arc length") of the fusing cell sheets is resolved long before fusion and that small differences in arc cell number are tolerated at fusion. The authors conclude that precision evolves over time and because arc length matches before cell number, suggest that two processes lead to these emergent properties. Such differences in arc cell number are accommodated by cell shape changes that are modulated by cell tension. When tension is perturbed by the expression of dominant negative myosin, matching is also perturbed. The authors also have documented changes in the structure of the formed seam with time, a process they call remodeling, and show that an initially flat interface between the zipped cell sheets becomes interlocked. Interestingly, the process of interlocking is accompanied by a loss of cadherin, actin and myosin at the interface between cells that came from contralateral sides of the embryo. The decrease in myosin occurs concomitantly with an increase in Bazooka/Par3, whereas patterns of vinculin accumulation is more similar to myosin. The authors conclude that such patterns of myosin and Bazooka accumulation are consistent with reduced tension in the formed seam. As described below, I do not understand how the observations that accompany attempts to reduce tension in stripes influence junctional remodeling, at least in part because I find Figure 6 very hard to understand and I find comparing the heat maps in Figure 7 hard to compare to the presentation of data in the rest of the Figures.

Overall, this is an interesting topic and a revised manuscript would make a compelling publication in *eLife*. My sense is that modifications to the presentation, and few if any new experiments, will make it ready for publication.

Specific comments:

The only way the authors could do these experiments were through live imaging and the data set they present is rich and valuable. Nevertheless, such live imaging requires the expression of fluorescently tagged cadherin, to label cell junctions and GFP-actin to delineate stripes vs. interstripes. Can the authors be sure that the expression of these transgenes doesn't affect the processes they are studying? Perhaps analysis of fixed and stained, control and experimental embryos could confirm that at least the overall pattern of mismatching, segment and intersegment widths and cell numbers is preserved? The analysis I envisage is to replicate the measurements on several fixed embryos stained with antibodies to cadherin and engrailed, to confirm that the patterns are not distorted in control vs. experimental animals.

Critical reading of the Ducuing and Vincent, (2016) and Pasakarnis et al., (2016) papers suggest that the authors of both studies have over interpreted their data to rule out a contribution of actin-myosin rich cables to wild type closure. Numerous other studies show that the cables are not strictly required for closure, but nevertheless normally contribute to the process. Similarly, the cables present but not necessary for wound healing. This does not mean that they do not normally facilitate the process. The authors should amend their Introduction appropriately. Indeed, the authors own observations that show expressing dominant negative myosin constructs in epithelial stripes slows closure is consistent with (but does not prove) a role for the purse strings in closure.

It is not clear to me why that authors normalize the difference in upper and lower arc lengths using Norm. ΔArc Length = (L_u_ – L_l_)/(L_u_ + L_l_). Shouldn't the ΔArc Length be normalized with the length of one arc, not two? That is, it would seem that (L_u_ – L_l_)/((L_u_ + L_l_)/2) is a more appropriate measure of the normalized arc length difference. A similar issue arises from the calculation of Norm. ΔWidth and with DME cell number. I think equations 4 and 5 are more appropriate normalization schemes. It seems that the authors' normalization scheme in equations 1, 2 and 3 underestimates all differences by two fold.

Shouldn't the authors be evaluating standard deviation rather than standard error of the mean? I'm not an expert on statistics, but see Barde and Barde, 2012, Perspectives in Clinical Research 3: 113.

Subsection “Population analysis of precision descriptors” states: "(control in Figures S3E and S3F)". I don't understand how the data in the two cited panels, which pertain to embryos expressing a dominant negative form of non muscle myosin constitute a "control". The authors should explain.

Subsection “Population analysis of precision descriptors”: there is an "are" before "remain" that should be deleted.

Subsection “Cell addition to the leading edge contributes to cell number and width equalization”. The authors conclude "This suggests that cell addition may be deployed to equalize number or width of the posterior compartment." I don't see how, taken at face value, that sentence is consistent with the observations reported in the rest of the paragraph. At the very least, it is misleading since cell addition occurs even when no disparity exists (previous to addition). The wording here should be amended to better reflect the observations reported in the remainder of the paragraph and accurately reflect a consistent interpretation of the data.

Subsection “Subsection “Cytoskeletal tension contributes to the evolution of precision and prevents segment misalignment”. On a sentence that starts with "Furthermore…" and focuses on the role of tension in cell matching, the authors describe the effects of the expression of dominant negative myosin in stripes and note that in one quarter of embryos there were fusions with "non-partner contralateral stripes". Such non-partner fusions were described by Franke et al., 2005 in embryos that had compromised myosin function (see their Figure 7). The authors should cite that work.

I am having trouble understanding how the experiments described in subsection "Bazooka and tension-dependent cell interlocking contributes to establishment of a continuous epithelial sheet" fits with their model of how tension influences remodeling of the formed seam. The authors show that cadherin, myosin, actin and vinculin are all decreased at the interface of the epidermal cell sheets from opposite flanks once the flanks are zipped. Moreover, they show that Bazooka is increased concomitantly with the decrease in myosin. They argue that tension is also decreased, and this allows an initially straight seam to become zig-zagged or interlocked. But here, they attempt to reduce tension by expressing either dominant negative myosin heavy chain or non phosphorylatable light chain and this disturbs remodeling. Shouldn't it do the opposite (i.e., like the over expression of Bazooka, described later in subsection "Bazooka and tension-dependent cell interlocking contributes to establishment of a continuous epithelial sheet")?

I find Figure 6 very confusing. First, D – F are labeled "closure" and G-I are labeled post closure, but each panel has a label "zipping" with an arrow. Isn't all zipping done post closure? Similarly, panels A, B and C have cartoons that are sketches of the interfaces seen in the micrographs and they too are labeled with zipping and arrows. But those arrows, in the control panel A are beneath mature interlocked cells, whereas the middle has T or + boundaries that show cells early in fusion. The authors clearly understand what they are trying to show. They need to figure out how to modify the Figure so it is more easily interpretable by the reader. I find the magnified inserts and schematics very helpful in A, B and C (but NOT the zipping arrows, which I don't fully understand). Why not have all of the panels comparable to A, B and C? I don't understand the point of showing zipping in two directions in A, B and C. The authors describe no differences between the anterior and posterior seams formed, so why bother showing both? The layout of panels, D – F is compelling because the dorsal opening is shown so it helps the reader orient the panel properly. Why not do that for A, B and C? Finally, are each pair D and G, E and H and F and I from the same embryo? If not, why not?

The authors wait until the Discussion section to tell us what the significance of a truncated form of Baz is. They add a sentence in the results to inform the reader.

I don't understand why the authors have switched to a heat map presentation in Figure 7. Of course, heat maps are useful for highlighting relative intensities, but the bulk of the rest of the paper depends on inverse fluorescent gray scale to depict cellular morphologies. So that the reader can better understand the effects of the various permutations of baz expression, the authors should present the data as they have for the other Figures. If heat maps are important, put them in a supplemental figure.

[Editors’ note: what now follows is the decision letter after the authors submitted for further consideration.]

Thank you for submitting your revised article "Cytoskeletal tension and Par3 tune interface geometry to ensure fusion fidelity and sheet integrity in dorsal closure" for consideration by *eLife*. Your article has been reviewed by three peer reviewers, one of whom is a member of our Board of Reviewing Editors, and the evaluation has been overseen by Anna Akhmanova as the Senior Editor. The reviewers have opted to remain anonymous.

The reviewers have discussed the reviews with one another and the Reviewing Editor has drafted this decision to help you prepare a revised submission.

This is a potentially interesting paper that addresses cell matching during the process of dorsal closure. It is from an accomplished lab that has contributed key findings to the field. The authors show that initially, the lateral epidermal cells that surround the dorsal opening are different in number and the perimeter of the dorsal opening measured on the right side of the embryo is different than the left side. The embryo adjusts length earlier than cell number and fusion between advancing cells sheets can occur even if the number of cells is not perfectly matched. The authors go on to show that an initially T or + seam that forms becomes interlocked and jagged. They go on to explore what might be responsible for regulating the process and investigate both tension (by perturbing myosin function and with laser cuts) and by manipulating Baz/Par3. They find effects with both, some of which are new and some of which are not (for example, the finding that DNZip causes segment misalignment). Overall, this is a very careful analysis, and the data add novel and important aspects to our understanding of epithelial fusion, showing that precision is not pre-set, but emerges.

This revised version of the manuscript is a major improvement to the manuscript submitted previously. The manuscript is easier to read, since they have put quite a few data into the Supplementary material. The figures are also significantly de-cluttered and it is now easier to digest the main points. We also appreciate the authors' efforts to address our point regarding better spatiotemporal control of their myosin experiments – we agree there is no easy way to do this without the development of new tools (ie. optogenetics as they point out) – and we concede that the alternative Gal4 driver (LE-Gal4) is a good compromise.

However, although the reviewers agree that the manuscript is easier to read, there are some concerns summarised below:

- If the authors could justify the assertions they summarize in their manuscript's Abstract, the paper would be a very nice contribution. Unfortunately, it is not clear that the assertions made are justified based on the data presented. Furthermore, the writing can still be improved. The reader is still left with a lot of detailed data. In order not to get lost, it would be helpful to add a summary sentence after each paragraph, if possible.

- The new Figure 5 and Figure 6 are still very difficult to follow. Is it possible to reduce the number of panels or perturbations in the main figures and move the remainder to supplementary figures? Some of the difficulty arises from the lengthy labels, e.g., "enGal4>UAS GFP," "enGal4>UAS ZipDN," "enGal4>UAS BazGFP ∆969-1464." Figure 5H is still rather confusing, in particular due to the inclusion of the significance within the columns. The authors may consider a modification of this graph, perhaps by showing three individual columns for each genotype, with the significance on top. Can the authors come up with more compact and understandable ways to label the different scenarios?

- A very interesting finding that is not well, if at all, highlighted is when equalization occurs. If the reviewers understand Figure 1—figure supplement 1Q correctly, don't the data suggest "action at a distance"? If geometry equalization occurs so early and number equalization occurs later, but still there is tens of microns between advancing cell sheets, how does one row of DME cells "know" to adjust its length and/or? Please clarify.

[Editors’ note: what now follows is the decision letter after the authors submitted for further consideration.]

Thank you for resubmitting your work entitled "Cytoskeletal tension and Bazooka tune interface geometry to ensure fusion fidelity and sheet integrity in dorsal closure" for further consideration at *eLife*. Your revised article has been favorably evaluated by Anna Akhmanova (Senior Editor), a Reviewing Editor, and one reviewer.

The manuscript has been improved but there are some remaining issues that need to be addressed before acceptance, as outlined below in reviewer #2's comments. As this version is the second round of submission, at this point, we would encourage you to hire a professional scientific editor to help clean up the manuscript. Otherwise the hard work that you have put into your experiments and analyses becomes obscured, which would be a great loss to the readership of *eLife*.

Reviewer #2:

I greatly appreciate the changes the authors have made to restructure the presentation of data, in particular by moving some of the data to supplementary material. However, before acceptance to *eLife*, this manuscript's writing still needs significant work. Many sentences contain vague modifiers. As one of the reviewers pointed out, each paragraph would benefit from a clear topic sentence that segues from the previous paragraph and introduces the main point of the new paragraph. For clarity, many sentences with multiple clauses could be made into multiple separate sentences.

General comment: the errors in measurements should be reported to the same decimal place as the value reported, not 11.81 {plus minus} 1.2 but rather 11.8 {plus minus} 1.2 (subsection “Bazooka modulates interface remodeling and DME cell dynamics during fusion” and other places).

Regarding the Discussion section, some of the comments are unfounded, such as "thermodynamically stable junctional configuration" (subsection “Fidelity in epithelial fusion: the logic and the mechanisms”), especially without a citation.

In the Discussion section, it is not clear to me what mechanisms are "emergent" and what the authors mean by this word. I suggest scaling back the claim of emergent mechanisms.

Sometimes the authors write "E Cadherin" or "E-cadherin" or "Ecadherin." These inconsistencies are a result of sloppy editing.

Throughout the manuscript, the authors write "this" without identifying a noun that follows that word. Such sentence structure is sloppy, as it is rarely clear which previous ideas the authors intend the word "this" to mean. A simple search for "this" and adding a noun (e.g., "observation," "result," "concept," etc.) would strengthen the clarity of these sentences.

I have outlined some example writing suggestions below, but this careful scrutiny needs to be applied to the entire manuscript, likely with the help of a professional editor.

Abstract:

- I believe the Abstract is still poorly written. For a broad journal like *eLife*, it should be accessible to a general life-sciences audience.

- Unclear writing: "We investigate this…": This what? Many ideas were presented previous to that sentence.

- "We find that although differences in cell number and length between the two fusing arcs reduce as fusion progresses, small differences in cell number between fusing epidermal segments persist at fusion" – this sentence is extremely confusing, and a general life-sciences audience will not know what the arcs are.

- The word "Rather" negates the previous sentence, which was also negated by the word "Although." This writing is unclear. Is it a positive result or a negative result that "small differences in cell number persist"?

- "Segment compartment boundaries" are not defined. "Front length" of what?

- "Emergent mechanisms" – it's not at all clear in the Abstract what the emergent mechanisms are.

Introduction:

- Even the first sentence is confusing. The comma after "healing" should not be there; the comma's presence forced me to read the sentence a number of times before I could follow what the authors were trying to say.

- The verb should be "has" because the subject of the sentence is the singular "study".

- The comma in” cell partners during *Drosophila* dorsal closure and eyelid closure, suggested that” is also incorrect and confusing, and the whole sentence should be rewritten for clarity.

- "It" is vague.

- What do the authors mean by "emergent"?

- This comma also is incorrect.

- At what time scale and in what process are the authors referring to "de novo"?

Results section:

- Is the list of measurements in subsection “Defining and measuring symmetry, fidelity and plasticity in fusion” missing the word "and" somewhere?

- Subsection “Defining and measuring symmetry, fidelity and plasticity in fusion”: what are "measures of plasticity in cell number"?

- The sentence in subsection “The emergence of symmetry and fidelity during dorsal closure” is confusing because of the use of "later" and "preceding" in reverse order. Can the authors rewrite this sentence more clearly?

- Subsection “The emergence of symmetry and fidelity during dorsal closure”: To what "mechanisms" are the authors referring?

- Subsection “Differential contributions of segment compartments to equalization in fusing front DME cell number and geometry”: better sentence structure would have the modifier "For this analysis" at the beginning of the sentence instead of the end.

- Subsection “Differential contributions of segment compartments to equalization in fusing front DME cell number and geometry”: what is a frequency of stripes? Do the authors mean number of stripes?

- Subsection “Differential contributions of segment compartments to equalization in fusing front DME cell number and geometry”: I don't understand the anthropomorphizing word "tolerated" here. What is tolerating one fewer cell?

- Subsection “Differential contributions of segment compartments to equalization in fusing front DME cell number and geometry”: comma needed after "pairs"; for sentences with two independent clauses separated by a conjunction, there needs to be a comma before the conjunction.

- Subsection “Differential contributions of segment compartments to equalization in fusing front DME cell number and geometry”: because of how confusingly written this paragraph is, the two different mechanisms are obscured.

- Subsection “Differential contributions of segment compartments to equalization in fusing front DME cell number and geometry”: "them" is an unclear pronoun.

- Subsection “Spatiotemporally regulated remodeling of cell junctions converts staggered en-face cellular contacts at the fusing front to an interlocking pattern after fusion”: what is "their" referring to?

- Subsection “Spatiotemporal regulation of tension and Bazooka at fusing interfaces”: missing a period.

- Subsection “Spatiotemporal regulation of tension and Bazooka at fusing interfaces”: since supplementary figures are possible in *eLife*, why are these data not shown?

- Subsection “Spatiotemporal regulation of tension and Bazooka at fusing interfaces”: why is this finding surprising?

- Subsection “Cytoskeletal tension entrains the evolution of precision and modulates cell behaviour and interface geometry”: which parameters?

- Subsection “Cytoskeletal tension entrains the evolution of precision and modulates cell behaviour and interface geometry”: should be "were" instead of "was".

- Subsection “Cytoskeletal tension entrains the evolution of precision and modulates cell behaviour and interface geometry”: what do the authors mean by "evolution"?

- Lines Results section, Discussion section: some letters are missing.

- Subsection “Bazooka modulates interface remodeling and DME cell dynamics during fusion”: whereas, not "where as".

---

## [Author Response]

[Editors’ note: the author responses to the first round of peer review follow.]

Reviewer #1:Dorsal closure in the Drosophila embryo has been used extensively as a model to unravel the genetic, molecular and biophysical principles of epithelial morphogenesis. During this process, the two lateral epidermal sheets move dorsalwards and eventually fuse at the dorsal midline. Up to now, very little is known about the fusion process, in particular with respect to the precision by which cells of both sides recognise each other in order to form a uniform epithelial sheet.In this manuscript, the authors set out to describe this process in great detail, focusing on the number and shape of the dorsalmost epidermal cells (DME) and the length covered by cells in stripes and interstripes, i. e. in the posterior and anterior compartments of the segments, respectively. Interestingly, they found that precision emerges over time, in particular with respect to the arc length, but that a certain degree of "imprecision" with respect to the cell number in contralateral stripes/interstripes is tolerated. Addition of cells from neighbouring compartments or from more ventral areas reduced the discrepancy in cell number in only some cases. They move on to show that the geometry of the junctions after fusion changes, from an "en-face" geometry to an "interlocked" pattern. This change is associated with a decrease of actin, Sqh and vinculin expression, being high in unfused cells, lower in en-face fused cells and even lower in "interlocked" cells. They further show that the proper change in junctional geometry depends on tension. As shown in other processes, e.g. germ band extension, expression of Bazooka (Baz)/Par3 shows an opposite behaviour, in that it is highest in "interlocked" cells. Upon overexpressing a truncated version of Baz, which is unable to localise at the membrane, this process is affected and goes along with reduced ECadherin expression, precocious interlocking, increased cell number disparity and changes in stripe width. Overall, most of the data presented here, which were obtained by an extremely careful analysis, add novel very important aspects for understanding epithelial fusion, showing that precision is not pre-set, but emerges.The experiments are carefully described, and the data are very well presented.

We thank the reviewer for these positive comments on the manuscript. Indeed, very little is known about whether and how precise cell matching occurs between the fusing flanks. Our work shows for the first time that while segment/compartment boundaries are respected, small differences in cell number between the fusing compartments is well tolerated. This begs an understanding of the nature of contacts formed upon fusion and motivated us to uncover the morphological/geometric and molecular changes that accompany it. Our work demonstrates for the first time that the contacts formed upon fusion remodel to generate an interlocked cell packing geometry at the dorsal midline. We also identified Bazooka and cytoskeletal tension as regulators of cell morphology, cell dynamics and interface remodelling. Both are themselves spatiotemporally regulated during the process, in a reciprocal manner.

However, I have a problem to follow the conclusion made on the role of Baz in this process. From the experiments presented in Figure 5E-5G they suggested that "Bazooka enrichment is associated with reduced interfacial tension". To corroborate this assumption, they first showed that impaired tension (achieved by expression ZipDN or SqhAA) affects junction remodelling. Do the authors have any idea why dominant negative Zip and Sqh have different effects on the process?

This is an interesting question that we also pondered about. ZipDN has stronger effects on epithelial order but milder effects on interlocking. In contrast, SqhAA has stronger effects on interlocking and milder effects on epithelial order. One explanation is that perturbing actinmyosin interactions (ZipDN) and perturbing actomyosin contractility by reducing phosphorylation on the myosin RLC (SqhAA) produce qualitatively distinct effects. An alternate possibility is that as both transgenes are expressed in otherwise wildtype backgrounds, their effects are different on account of the varying stoichiometries of wildtype and mutant myosins in each perturbation. At present, we cannot distinguish between these two possibilities. A strict assessment of the reason behind the difference will require an analysis of their relative stoichiometries and the ability to target the mutants solely to the fusing interfaces at different times during the process. In the new data we provide using the leading edge Gal4 (Figure 5—figure supplement 1), we show that a phosphomimetic myosin RLC mutant (SqhEE, that enhances contractility) also effectively abolishes interlocking and renders the midline seam ramrod straight. In contrast, the other two perturbations resulted in a wavy dorsal midline and produced patches of defective interlocking. This local perturbation (in contrast to the expression in stripes) lends support to the idea that the downregulation of contractility in the leading edge cell fusing interfaces is necessary for their relaxation and remodeling. It also brings out the importance of the spatiotemporal regulation of actomyosin based contractility (see also response to reviewer 3).

They go on to express either GFP tagged full-length Baz, a GFP tagged C-terminally truncated Baz (Baz[Δ969-1464]) or Baz-RNAi. Expression of either full-length Baz or Baz[Δ969-1464] resulted in similar phenotypes, including weaker ECadherin staining, whereas expression of Baz-RNAi had only mild effects, but showing increased E-Cad staining. Baz-GFP was recruited with a similar dynamics as endogenously-tagged Baz. In contrast, Baz[Δ969-1464] recruitment was delayed, which was associated with an extended time required to reach the interlocked stage. Finally, they induced a wound in the epidermis on one side of the embryo and observed the fusion event in the wound. From all this they conclude that "… changes in interface length and geometry are dependent on regulation of cytoskeletal tension and Baz/Par3" (Abstract). While the data are convincing with respect to the importance of cytoskeletal tension in this process, the role of Baz is unclear. The fact that overexpression of Baz versions, in particular the truncated one, impairs this process, does not imply that Baz plays a role in this process, given that expression of RNAi has a different outcome. Rather, expressing this form results in a dominant phenotype, but does not provide convincing evidence on the role of Baz in this process.

In the earlier version of the manuscript, we had analysed the effects of Bazooka using three perturbations targeted to the stripe: full length Bazooka overexpression, overexpression of a truncated version of Bazooka and Bazooka RNAi. The phenotypes of the former two were assessed both with live imaging (GFP tagged constructs) and in fixed preparations. The latter was only analysed in fixed preps (with UAS GFP/engrailed antibody to mark the engrailed domain). In the present version, we have examined Bazooka RNAi embryos using fixed preparations in greater detail (Figure 5, Figure 5—figure supplement 1) and also in real time using sqh::Utrophin GFP (Figure 6, Figure 7). We chose Utrophin GFP as it is neutral (binds actin) and is unlikely to modify the phenotype of Bazooka RNAi, a fear we had with using ECadherin GFP or sqh GFP. We also analysed epithelial order and junction remodeling in fixed preparations as well as in real time. Our new analysis reveals that the expression of either Baz RNAi or BazΔ in stripes produces a significant increase in the percentage of fused interfaces that were curved/squiggly compared to controls, and a significant decrease in both angled and straight interfaces. In contrast Baz overexpression only modestly reduced the proportion of straight interfaces (Figure 5). In Baz RNAi and BazΔ expressing stripes, cells in the stripe appeared more “fluid” and mobile and transgressed the dorsal midline leading to an obliterated seam, suggesting that Bazooka restrains/ regulates cell dynamics and ensures epithelial order at the midline (Figure 5). In contrast Bazooka overexpression led to the ordered stacking of cell rows and increased order at the dorsal midline and reduced cell shape anisotropy (squarish rather than rectangular cells; Figure 5,Figure 5—figure supplement 1G and Video 8A,B).

We also reexamined the effect of the three perturbations on the levels of E cadherin. Both Baz overexpression transgenes reduced the intensity of Ecadherin throughout the stripe. The most striking effect in Baz RNAi and BazΔ was a reduction in the anisotropy in the distribution of Ecadherin in DME cells. This was not observed upon Baz overexpression. In 15% of Baz RNAi expressing stripes, an enrichment of E Cadherin specifically at the fusing interfaces was observed even after fusion. The mobility of the interfaces we are examining makes it difficult to determine whether Bazooka influences the relative distributions of mobile and immobile ECadherin at these interfaces (Figure 5).

We have now examined the effect of Baz RNAi on actin remodeling during fusion in real time and find that the clearing of actin from the fusing interfaces after fusion is impaired in the absence of Baz (Figure 6, Figure 7, Video 10, Video 11). In Baz RNAi, actin persists for longer at these interfaces. Thus, our view of Bazooka function based on the effects of RNAi is that Bazooka restrains cell movement within a sheet and negatively regulates the actomyosin contractility at the fusing interfaces. The temporally regulated appearance of Bazooka at the fusing interfaces thus ensures junction relaxation and stabilization of adhesion in the midline. Our results do not rule out a role for Bazooka dependent ECadherin regulation. Precisely how Bazooka influences junction and cytoskeletal remodeling is an ongoing project in the lab.

Reviewer #2:This manuscript by Das Gupta and Narasimha employs an elegant combination of genetics, live imaging, and morphometric analyses to study re-epithelialization in Drosophila dorsal closure. The authors first perform a detailed descriptive analysis of segment matching in terms of cell number and cell shape and uncover a role for myosin regulation (or, in their terms, tension) in this process. The authors then explore the resolution of junctional geometry following fusion. The results suggest that reciprocal regulation of Bazooka/Par-3 and vinculin localization downstream of cytoskeletal tension are required to effectively restore epithelial integrity.While the study includes an impressive volume of detailed, quantitative analyses, the manuscript suffers from poor organization and lack of structure, which makes it challenging for the reader to tease apart the real significance of the authors' contributions. The finding that cell numbers between opposing sides of the embryo are not initially precisely regulated is interesting, as are the authors' data on reorganization of junctions following fusion. However, these two components of the story have not been well melded together, and it is difficult for the reader to see the forest for the trees. Despite these qualms, the videos are stunning, and the experiments are, for the most part, well controlled and carefully presented in the figures and captions-too much so at times, in terms of the amount of detail presented.

We thank you for your valuable comments. In the revised version of the manuscript, we have both rewritten and reorganized the sections (and therefore figures) to ensure better clarity, readability and better melding of the two parts.

Major concerns:1) In the Introduction and Discussion section, the authors could do a better job explaining how the aspects of dorsal closure they are studying might have implications beyond this particular system. Concepts key to the authors' work, such as fidelity and precision, as they apply to this particular system, should be more clearly defined in the Introduction.

We have expanded the Introduction to include our definitions of precision/fidelity, highlighted the need to understand the underlying mechanisms both from the perspective of how molecular and morphological symmetry is ensured during large-scale cell movements and also from the perspective of reepithelialisation. We have also posed alternative hypotheses of how this might be achieved.

2) The authors perturb myosin as a way to make inferences about the role of tissue tension in this process. However, their strategy is somewhat problematic, as it relies on constitutive expression of a DN-zipper construct in engrailed stripes. While it is likely that tension does play a role in this process, it is difficult to interpret whether the observed changes in precision parameters occur because of a direct role for actomyosin-based tension in cellular rearrangements (which I believe that the authors want to imply) or that other aspects such as cell proliferation might be impaired. To address this problem, the authors should treat embryos with ROCK inhibitor Y-27632 during dorsal closure, which will allow more temporal control over manipulation of contractility.

We agree that our perturbations targeted to the engrailed compartment do not allow us to confine them temporally and spatially to the period of dorsal closure during which time neither the DME nor the amnioserosa cells divide. We have also confirmed this in our experiments using UAS histone RFP that allows us to see mitotic figures and watch divisions as they happen. The use of drugs is also problematic: they do not allow the spatial control of contractility as they cannot be injected specifically into the relevant cells. The amnioserosa will also be affected and this can influence the phenotypes we see. Given the spatiotemporal dynamics of actomyosin we see, and the fact that different stripes in the same embryo are in different stages of fusion at any given stage, the ideal way of addressing this is question is to use optogenetics to trigger and target the perturbations to single interfaces. This is currently not possible. Another compounding feature is that changes in cytoskeletal tension regulate more than just cell rearrangements: they influence junction remodeling at fusing interfaces and affect dorsal closure dynamics. We therefore reasoned that a good compromise would be to drive the three perturbations that alter tension, ZipDN, SqhAA and SqhEE exclusively in the leading edge cells using the leading edge Gal4 whose expression levels increase and patchiness decreases as dorsal closure progresses. While the patchiness does not allow us to assess fidelity at the level of each compartment, we find effects on straightness of the dorsal midline that suggest reduced fidelity (Figure 5—figure supplement 1). In both ZipDN and SqhAA, the dorsal midline is wavy along the AP axis where as in SqhEE, the midline is ramrod straight. In all three perturbations, and in particular, in SqhEE, poor interlocking is observed in post-closure embryos. These experiments have allowed us to confirm that altering contractility in just the leading edge cells during dorsal closure (at stages when proliferation is complete) influences junction remodeling, fusion fidelity and epithelial order at the dorsal midline.

3) As written, the manuscript is largely descriptive and difficult to digest. The novel points of the paper, although beautiful, are subtle and might be lost on readers unfamiliar with the extensive literature on dorsal closure. In its current form, this study may be better suited for a more specialist audience than the readers of eLife. This outcome would be a shame because the problem of re-epithelialization is of broad interest, and the authors' observations regarding interface remodelling after fusion are interesting, but they are regrettably buried in a sea of difficult-to-follow quantifications of unclear broader biological significance.

We have incorporated these suggestions in rewriting this version of the manuscript. We have now moved quantifications that are not essential to the supplementary materials along the lines suggested in 4. We hope you will agree that the most important results now stand out.

4) To begin to unclutter the story, Figure 1 and Figure 2, a tour de force of measurements and quantifications, could be simplified, with many of the detailed comparisons between stripes and interstripes removed or moved to supplemental. Similarly, Figure 7 and Figure 2—figure supplement 1 could be presented with fewer time points, just those illustrating the authors points. It is hard for the reader to digest so many panels.

Thank you for this suggestion. We have incorporated these changes in the in the revised figures.

Reviewer #3:This manuscript focuses on the important problem of fusion fidelity and seamless epithelial continuity in the dorsal closure model for epithelial cell sheet migration and fusion. The authors find that differences in the length of the leading edge (here referred to as "arc length") of the fusing cell sheets is resolved long before fusion and that small differences in arc cell number are tolerated at fusion. The authors conclude that precision evolves over time and because arc length matches before cell number, suggest that two processes lead to these emergent properties. Such differences in arc cell number are accommodated by cell shape changes that are modulated by cell tension. When tension is perturbed by the expression of dominant negative myosin, matching is also perturbed. The authors also have documented changes in the structure of the formed seam with time, a process they call remodeling, and show that an initially flat interface between the zipped cell sheets becomes interlocked. Interestingly, the process of interlocking is accompanied by a loss of cadherin, actin and myosin at the interface between cells that came from contralateral sides of the embryo. The decrease in myosin occurs concomitantly with an increase in Bazooka/Par3, whereas patterns of vinculin accumulation is more similar to myosin. The authors conclude that such patterns of myosin and Bazooka accumulation are consistent with reduced tension in the formed seam. As described below, I do not understand how the observations that accompany attempts to reduce tension in stripes influence junctional remodeling, at least in part because I find Figure 6 very hard to understand and I find comparing the heat maps in Figure 7 hard to compare to the presentation of data in the rest of the Figures.Overall, this is an interesting topic and a revised manuscript would make a compelling publication in eLife. My sense is that modifications to the presentation, and few if any new experiments, will make it ready for publication.

We thank the reviewer for the positive comments.

Specific comments:The only way the authors could do these experiments were through live imaging and the data set they present is rich and valuable. Nevertheless, such live imaging requires the expression of fluorescently tagged cadherin, to label cell junctions and GFP-actin to delineate stripes vs. interstripes. Can the authors be sure that the expression of these transgenes doesn't affect the processes they are studying? Perhaps analysis of fixed and stained, control and experimental embryos could confirm that at least the overall pattern of mismatching, segment and intersegment widths and cell numbers is preserved? The analysis I envisage is to replicate the measurements on several fixed embryos stained with antibodies to cadherin and engrailed, to confirm that the patterns are not distorted in control vs. experimental animals.

We had already done this. In one set of experiments in which we were examining the effect of Bazooka RNAi (presented in the earlier version of the manuscript), we used the engrailed antibody to mark the engrailed domain in controls and mutants, and Ecadherin antibody to label cell outlines, with no additional transgenes. We did this also to see whether the addition of UAS GFP/RFP to label the compartment may have reduced phenotypic severity. A second set of controls used UAS GFP (neutral) to mark stripes in controls and mutants (controls for UAS Baz RNAi, UAS SqhAA, UAS SqhEE) in which the stripes were labelled with UAS GFP and cell boundaries with Ecadherin antibody. Although a dynamic analysis of the parameters could not done in these two data sets, DME cell numbers and their differences, compartment widths and interface geometry were comparable in both sets. For the real time analysis, we used two control genotypes: i) Ecadherin GFP with en>UAS actin GFP (Zip DN experiments) and ii) sqh::Utrophin GFP (also neutral in that it expresses a GFP that can bind actin) with en>UAS NLS RFP (UAS Baz RNAi experiments). In each case, the mutant and controls both had ECadherin GFP or sqh::UtrGFP. Furthermore, the two control genotypes yielded similar results for DME cell numbers and cell number difference evolution, stripe widths as well as for the timescales for interface remodelling suggesting that the parameters we are measuring are not influenced by the transgenes used for live imaging. Specifically, in this manuscript, we show (in Figures 6F,G) that the values of the differences in DME cell number and constriction index (stripe width/stripe cell number) are in the same range in two different controls used: ECadhGFP with en>UAS actin GFP and sqh::Utrophin GFP with en>UAS NLS RFP.

Critical reading of the Ducuing and Vincent, (2016) and Pasakarnis et al., (2016) papers suggest that the authors of both studies have over interpreted their data to rule out a contribution of actin-myosin rich cables to wild type closure. Numerous other studies show that the cables are not strictly required for closure, but nevertheless normally contribute to the process. Similarly, the cables present but not necessary for wound healing. This does not mean that they do not normally facilitate the process. The authors should amend their Introduction appropriately. Indeed, the authors own observations that show expressing dominant negative myosin constructs in epithelial stripes slows closure is consistent with (but does not prove) a role for the purse strings in closure.

We fully agree with this view and have amended our statement in the Introduction and also included a statement in the Discussion section. Indeed, our new results (Figure 5—figure supplement 1) with leading edge Gal4 that is expressed only in the DME cells also confirms the view that tension generated in the leading edge cell entrains the spatiotemporal dynamics of dorsal closure.

It is not clear to me why that authors normalize the difference in upper and lower arc lengths using Norm. ΔArc Length = (L_u_ – L_l_)/(L_u_ + L_l_). Shouldn't the ΔArc Length be normalize with the length of one arc, not two? That is, it would seem that (L_u_ – L_l_)/((L_u_ + L_l_)/2) is a more appropriate measure of the normalized arc length difference. A similar issue arises from the calculation of Norm. ΔWidth and with DME cell number. I think equations 4 and 5 are more appropriate normalization schemes. It seems that the authors' normalization scheme in equations 1, 2 and 3 underestimates all differences by two fold.

We have now incorporated these suggestions in the new Graphs presented, and in the Materials and methods section.

Shouldn't the authors be evaluating standard deviation rather than standard error of the mean? I'm not an expert on statistics, but see Barde and Barde, 2012, Perspectives in Clinical Research 3: 113.

In the majority of the data presented (except frequency/percentage distributions of data values), the graphs show all the data points, the horizontal bars indicate the median and interquartile range, and the Mann-Whitney test was performed to detect significant differences in the shape and spread of the distribution as well as differences in the medians between the genotypes/conditions being compared. The current Figure 1H is also now represented in this way rather than as a histogram with mean and SEM. We have used standard deviation where we were keen to determine the data spread/scatter/variability. We have replaced the error bars in the current Figure 4C to be standard deviations rather than SEM. In some graphs in which we are expressing differences in relative frequency (eg: Figure 5H and Figure 1—figure supplement 3F), we have used the SEM (which relies on the SD and the sample size) as we were interested in determining the accuracy of the mean.

Subsection “Population analysis of precision descriptors” states: "(control in Figures S3E and S3F)". I don't understand how the data in the two cited panels, which pertain to embryos expressing a dominant negative form of non muscle myosin constitute a "control". The authors should explain.

We should have said control panels in Figure 1—figure supplement 3E and Figure 1—figure supplement 3F. The control we were referring to is not Zip DN but to the controls presented on the left hand side of each graph (ECadherin GFP with en>UAS actin GFP), for comparison.

Subsection “Population analysis of precision descriptors”: there is an "are" before "remain" that should be deleted.

This has been corrected.

Subsection “Cell addition to the leading edge contributes to cell number and width equalization”. The authors conclude "This suggests that cell addition may be deployed to equalize number or width of the posterior compartment." I don't see how, taken at face value, that sentence is consistent with the observations reported in the rest of the paragraph. At the very least, it is misleading since cell addition occurs even when no disparity exists (previous to addition). The wording here should be amended to better reflect the observations reported in the remainder of the paragraph and accurately reflect a consistent interpretation of the data.

We agree. What we see is that in a small percentage of stripes, the cell addition reduces cell number disparity but in the majority, there is a correlation between width equalization and cell addition. We have corrected this to reflect what our observations indicate.

Subsection “Cytoskeletal tension contributes to the evolution of precision and prevents segment misalignment”. On a sentence that starts with "Furthermore…" and focuses on the role of tension in cell matching, the authors describe the effects of the expression of dominant negative myosin in stripes and note that in one quarter of embryos there were fusions with "non-partner contralateral stripes". Such non-partner fusions were described by Franke et al., 2005 in embryos that had compromised myosin function (see their Figure 7). The authors should cite that work.

We thank you for pointing this out to us. We have now cited this work.

I am having trouble understanding how the experiments described in subsection "Bazooka and tension-dependent cell interlocking contributes to establishment of a continuous epithelial sheet" fits with their model of how tension influences remodeling of the formed seam. The authors show that cadherin, myosin, actin and vinculin are all decreased at the interface of the epidermal cell sheets from opposite flanks once the flanks are zipped. Moreover, they show that Bazooka is increased concomitantly with the decrease in myosin. They argue that tension is also decreased, and this allows an initially straight seam to become zig-zagged or interlocked. But here, they attempt to reduce tension by expressing either dominant negative myosin heavy chain or non phosphorylatable light chain and this disturbs remodeling. Shouldn't it do the opposite (i.e., like the over expression of Bazooka, described later in subsection "Bazooka and tension-dependent cell interlocking contributes to establishment of a continuous epithelial sheet")?

We understand your concern. In the experiments presented we perturb tension in the entire DME cell and in the entire engrailed compartment. We however show that tension is anisotropic. Thus, although tension is normally reduced in the fusing interfaces, in our perturbations, it is also abolished in the non-fusing interfaces, as for instance the DV oriented interfaces of the leading edge cells along which it is positioned and potentially also in the apicomedial pool. In both germ band extension and pulsed constriction in the amnioserosa, it has been shown that junction relaxation relies on directed actomyosin contractility that pulls the vertices outward. In our context, the neighbours could be DME cells in the same flank or in the opposite flank, where cytoskeletal tension may need to be high. Alternatively, tension in the compartment boundaries may also have an effect. Our results with LE Gal4 (which is patchy) suggest that perturbing cytoskeletal tension only in the leading edge cells is sufficient to influence both interface remodeling and fusion fidelity. Indeed, in the work we present in Figure 5—figure supplement 1, we show that all affect interlocking in patches, with SqhEE having the strongest effect. We believe that we have now established a nice model system in which can address autonomy, non-autonomy and the impact of tension anisotropies within the DME cells in real time.

I find Figure 6 very confusing. First, D – F are labeled "closure" and G-I are labeled post closure, but each panel has a label "zipping" with an arrow. Isn't all zipping done post closure? Similarly, panels A, B and C have cartoons that are sketches of the interfaces seen in the micrographs and they too are labeled with zipping and arrows. But those arrows, in the control panel A are beneath mature interlocked cells, whereas the middle has T or + boundaries that show cells early in fusion. The authors clearly understand what they are trying to show. They need to figure out how to modify the Figure so it is more easily interpretable by the reader. I find the magnified inserts and schematics very helpful in A, B and C (but NOT the zipping arrows, which I don't fully understand). Why not have all of the panels comparable to A, B and C? I don't understand the point of showing zipping in two directions in A, B and C. The authors describe no differences between the anterior and posterior seams formed, so why bother showing both? The layout of panels, D – F is compelling because the dorsal opening is shown so it helps the reader orient the panel properly. Why not do that for A, B and C? Finally, are each pair D and G, E and H and F and I from the same embryo? If not, why not?

We understand the confusion. This was a semantic error: zipping was wrongly used to indicate the direction of fusion (from the canthi inwards). We simply wanted to indicate that the segments at the anterior and posterior ends of the ellipse fuse first while the central segments fuse last and that remodeled interfaces are found outwards from the canthi while remodeling interfaces are found closer to the canthi. We have now removed “zipping”and “remodeling” from the figures and instead indicated the position of the midpoint along the AP axis to orient the reader to region of the dorsal epidermis from which the image is obtained. The data in each pair are not from the same embryo because this data was obtained from fixed preparations of embryos that were either still completing closure (some amnioserosa still visible) or immediately after closure is complete (no amnioserosa visible on the dorsal surface).

The authors wait until the Discussion section to tell us what the significance of a truncated form of Baz is. They add a sentence in the results to inform the reader.

We have done this now.

I don't understand why the authors have switched to a heat map presentation in Figure 7. Of course, heat maps are useful for highlighting relative intensities, but the bulk of the rest of the paper depends on inverse fluorescent gray scale to depict cellular morphologies. So that the reader can better understand the effects of the various permutations of baz expression, the authors should present the data as they have for the other Figures. If heat maps are important, put them in a supplemental figure.

We have switched to the grey scale images and have used some heat maps where spatiotemporal regulation of intensity is what we want to show, in the supplementary figures.

[Editors' note: the author responses to the re-review follow.]

The reviewers have discussed the reviews with one another and the Reviewing Editor has drafted this decision to help you prepare a revised submission.This is a potentially interesting paper that addresses cell matching during the process of dorsal closure. It is from an accomplished lab that has contributed key findings to the field. The authors show that initially, the lateral epidermal cells that surround the dorsal opening are different in number and the perimeter of the dorsal opening measured on the right side of the embryo is different than the left side. The embryo adjusts length earlier than cell number and fusion between advancing cells sheets can occur even if the number of cells is not perfectly matched. The authors go on to show that an initially T or + seam that forms becomes interlocked and jagged. They go on to explore what might be responsible for regulating the process and investigate both tension (by perturbing myosin function and with laser cuts) and by manipulating Baz/Par3. They find effects with both, some of which are new and some of which are not (for example, the finding that DNZip causes segment misalignment). Overall, this is a very careful analysis, and the data add novel and important aspects to our understanding of epithelial fusion, showing that precision is not pre-set, but emerges.This revised version of the manuscript is a major improvement to the manuscript submitted previously. The manuscript is easier to read, since they have put quite a few data into the Supplementary material. The figures are also significantly de-cluttered and it is now easier to digest the main points. We also appreciate the authors' efforts to address our point regarding better spatiotemporal control of their myosin experiments – we agree there is no easy way to do this without the development of new tools (ie. optogenetics as they point out) – and we concede that the alternative Gal4 driver (LE-Gal4) is a good compromise.

We thank the reviewers for their positive comments on our revised manuscript. Our manuscript demonstrates for the first time that precision emerges during the process of fusion and identifies the cellular and molecular mechanisms that contribute to it. Although Zip DN has previously been shown junction remodeling events we describe here for the first time. We also demonstrate for the first time, the requirement of Bazooka for ensuring fusion fidelity and to enable junction remodeling. While the effects of some of the perturbations we have analysed may seem small yet significant, we believe from our examination of zygotic Bazooka mutant embryos that these are the result of incomplete loss of function. Indeed, Bazooka mutant embryos often fail in dorsal closure (new data added in this revision), precluding our ability to perform the kind of detailed analysis done on the other perturbations. The junctional morphologies evident in these mutants convince us that the effects are real and are stark. We also concede that there is variability in interface morphologies at fusion. Our quantitative analysis of remodeling in control embryos and in mutants captures this variability but also uncovers significant shifts in the frequency distribution of junctions of different morphologies. In this revised version of the manuscript, we present data that captures the differences in frequency distribution of interface orientation angles as well as the central tendencies of the three described categories of interfaces for all the Bazooka perturbations (Figures 5H-K) and for myosin perturbation SqhAA (now in Figure 5—figure supplement 1).

However, although the reviewers agree that the manuscript is easier to read, there are some concerns summarised below:- If the authors could justify the assertions they summarize in their manuscript's Abstract, the paper would be a very nice contribution. Unfortunately, it is not clear that the assertions made are justified based on the data presented. Furthermore, the writing can still be improved. The reader is still left with a lot of detailed data. In order not to get lost, it would be helpful to add a summary sentence after each paragraph, if possible.

We do believe that we have justified all the assertions in the Abstract: (i) that precision is not preset but evolves; (ii) that cell matching happens at less than perfect fidelity; (iii) that fusion happens despite differences in cell number; (iv) that changes in interface length and geometry ensures segment alignment and seamless continuity and (v) that tension generated by the actomyosin cytoskeleton and the polarity protein Bazooka modulate cellular rearrangements and changes in interface morphology that mediate precision and seamless continuity. We address the other concerns below.

- The new Figure 5 and Figure 6 are still very difficult to follow. Is it possible to reduce the number of panels or perturbations in the main figures and move the remainder to supplementary figures? Some of the difficulty arises from the lengthy labels, e.g., "enGal4>UAS GFP," "enGal4>UAS ZipDN," "enGal4>UAS BazGFP ∆969-1464." Figure 5H is still rather confusing, in particular due to the inclusion of the significance within the columns. The authors may consider a modification of this graph, perhaps by showing three individual columns for each genotype, with the significance on top. Can the authors come up with more compact and understandable ways to label the different scenarios?

We have split the erstwhile Figure 5H to include additional panels H-J and have now moved panels Figure 6 H,I to Figure 6—figure supplement 1. We have made the labels less long.

- A very interesting finding that is not well, if at all, highlighted is when equalization occurs. If the reviewers understand Figure 1—figure supplement 1Q correctly, don't the data suggest "action at a distance"? If geometry equalization occurs so early and number equalization occurs later, but still there is tens of microns between advancing cell sheets, how does one row of DME cells "know" to adjust its length and/or? Please clarify.

Yes. Our results in Figure 1—figure supplement 1Q reveal that mechanisms that contribute to equalization happen well before fusion and suggest action at a distance. We are currently investigating whether sensing operates across the amnioserosa or along the circumference of each segment compartment and whether this is mechanically or chemically regulated. Our results also show that although the mechanisms kick in well before fusion, final adjustments in compartment width after fusion ensure segment alignment in compartments in which there is still a stagger or width discrepancy. This suggests that while there are hardwired mechanisms that are deployed to ensure precision, their deployment is based on an as yet unidentified ‘sensing’ mechanism. The mechanisms that contribute to the evolution of precision are in some ways reminiscent of “kinetic proofreading” during translation. We have now discussed this in the revised text.

[Editors' note: the author responses to the re-review follow.]

Reviewer #2:I greatly appreciate the changes the authors have made to restructure the presentation of data, in particular by moving some of the data to supplementary material. However, before acceptance to eLife, this manuscript's writing still needs significant work. Many sentences contain vague modifiers. As one of the reviewers pointed out, each paragraph would benefit from a clear topic sentence that segues from the previous paragraph and introduces the main point of the new paragraph. For clarity, many sentences with multiple clauses could be made into multiple separate sentences.General comment: the errors in measurements should be reported to the same decimal place as the value reported, not 11.81 {plus minus} 1.2 but rather 11.8 {plus minus} 1.2 (subsection “Bazooka modulates interface remodeling and DME cell dynamics during fusion” and other places).

We have now corrected this throughout the manuscript and ensured that we use the same decimal place (single) for both the value and its error.

Regarding the Discussion section, some of the comments are unfounded, such as "thermodynamically stable junctional configuration" (subsection “Fidelity in epithelial fusion: the logic and the mechanisms”), especially without a citation.

We have now removed any mention of thermodynamic stability from the manuscript.

In the Discussion section, it is not clear to me what mechanisms are "emergent" and what the authors mean by this word. I suggest scaling back the claim of emergent mechanisms.

We have used the word “adaptive” rather than emergent to indicate mechanisms that are put into action as a response to the sensing of disparities and arise as a consequence of cell dynamics. It was also in this context that we had used the word “emergent”.

Sometimes the authors write "E Cadherin" or "E-cadherin" or "Ecadherin." These inconsistencies are a result of sloppy editing.

We have now corrected this throughout the manuscript and have ensured that we use the same notation throughout.

Throughout the manuscript, the authors write "this" without identifying a noun that follows that word. Such sentence structure is sloppy, as it is rarely clear which previous ideas the authors intend the word "this" to mean. A simple search for "this" and adding a noun (e.g., "observation," "result," "concept," etc.) would strengthen the clarity of these sentences.

We have now corrected this throughout the manuscript and ensured that a noun follows “This”.

I have outlined some example writing suggestions below, but this careful scrutiny needs to be applied to the entire manuscript, likely with the help of a professional editor.

We greatly appreciate your thorough reading of our manuscript and thank you for your comments and suggestions.

Abstract:- I believe the Abstract is still poorly written. For a broad journal like eLife, it should be accessible to a general life-sciences audience.

We have edited the manuscript in the hope that it will be easily understood by the broad readership of *eLife*.

- Unclear writing: "We investigate this…": This what? Many ideas were presented previous to that sentence.

We have removed this sentence from the Abstract now.

- "We find that although differences in cell number and length between the two fusing arcs reduce as fusion progresses, small differences in cell number between fusing epidermal segments persist at fusion" – this sentence is extremely confusing, and a general life-sciences audience will not know what the arcs are.

We have introduced flanks and segments in the Abstract now and do not refer to arcs. We define arcs in the Results section.

- The word "Rather" negates the previous sentence, which was also negated by the word "Although." This writing is unclear. Is it a positive result or a negative result that "small differences in cell number persist"?

We wanted to convey that small differences in cell number persist at fusion and that fidelity with respect to cell number matching is less than perfect (ie, the disparity in cell number between the fusing pairs is not always zero). We have now edited the Abstract so that neither word (although or rather) is used.

- "Segment compartment boundaries" are not defined. "Front length" of what?

We have introduced segmentation in the abstract now. We meant length or width of the fusing fronts of the flank or segment.

- "Emergent mechanisms" – it's not at all clear in the Abstract what the emergent mechanisms are.

We have removed “emergent” from the Abstract.

Introduction:- Even the first sentence is confusing. The comma after "healing" should not be there; the comma's presence forced me to read the sentence a number of times before I could follow what the authors were trying to say.

We have edited this sentence.

- The verb should be "has" because the subject of the sentence is the singular "study".

This has been corrected to “Earlier studies on epithelial fusion events … have identified”.

- The comma in “cell partners during Drosophila dorsal closure and eyelid closure, suggested that” is also incorrect and confusing, and the whole sentence should be rewritten for clarity.

This sentence has been corrected.

- "It" is vague.

This has been corrected. “It” has been replaced by “Dorsal closure”.

- What do the authors mean by "emergent"?

We have used the word “adaptive” rather than emergent to indicate mechanisms that are put into action as a response to the sensing of disparities that are arise as a consequence of cell dynamics. It was also in this context that we had used the word “emergent”.

- This comma also is incorrect.

This sentence has been corrected.

- At what time scale and in what process are the authors referring to "de novo"?

We used de novo to indicate that remodelling occurs during the formation of new junctions at fusion. We wanted to distinguish it from the remodeling of existing junctions as seen in germband extension. We now simply use “formation of new contacts” or “formation of the new seam” where de novo previously appeared in the text.

Results section:- Is the list of measurements in subsection “Defining and measuring symmetry, fidelity and plasticity in fusion” missing the word "and" somewhere?

It is indeed missing an “and”. We have corrected this.

- Subsection “Defining and measuring symmetry, fidelity and plasticity in fusion”: what are "measures of plasticity in cell number"?

We used “plasticity” to simply mean the ability to modulate cell number or geometry. We have corrected this sentence.

- The sentence in subsection “The emergence of symmetry and fidelity during dorsal closure” is confusing because of the use of "later" and "preceding" in reverse order. Can the authors rewrite this sentence more clearly?

We have changed this sentence now.

- Subsection “The emergence of symmetry and fidelity during dorsal closure”: To what "mechanisms" are the authors referring?

We suggest that distinct cellular mechanisms must enable differences in cell number and differences in the geometry within each fusing compartment.

- Subsection “Differential contributions of segment compartments to equalization in fusing front DME cell number and geometry”: better sentence structure would have the modifier "For this analysis" at the beginning of the sentence instead of the end.

We have changed this sentence now.

- Subsection “Differential contributions of segment compartments to equalization in fusing front DME cell number and geometry”: what is a frequency of stripes? Do the authors mean number of stripes?

As mentioned in Materials and methods section, stripes refer to the posterior compartment of the embryonic segment. We agree that frequency of stripes is confusing. We meant “proportion of stripes” of a given kind. We have now changed “frequency” to “proportion.”

- Subsection “Differential contributions of segment compartments to equalization in fusing front DME cell number and geometry”: I don't understand the anthropomorphizing word "tolerated" here. What is tolerating one fewer cell?

We have removed “tolerated”. We used it to imply that despite small differences in cell number (of one or two cells) in the fusing segment compartments, the compartment boundaries aligned. The sentence has been reworded.

- Subsection “Differential contributions of segment compartments to equalization in fusing front DME cell number and geometry”: comma needed after "pairs"; for sentences with two independent clauses separated by a conjunction, there needs to be a comma before the conjunction.

We have added this now.

- Subsection “Differential contributions of segment compartments to equalization in fusing front DME cell number and geometry”: because of how confusingly written this paragraph is, the two different mechanisms are obscured.

We have now split this section into paragraphs and added two new paragraphs with separate section headings to emphasise the two mechanisms we are referring to. Subsection “Bazooka modulates interface remodelling during fusion” and subsection “Bazooka entrains actomyosin contractility during epithelial fusion”.

- Subsection “Differential contributions of segment compartments to equalization in fusing front DME cell number and geometry”: "them" is an unclear pronoun.

We have edited this sentence to read “For this, we determined the variance in DME cell numbers between ipsilateral and contralateral stripe pairs”.

- Subsection “Spatiotemporally regulated remodeling of cell junctions converts staggered en-face cellular contacts at the fusing front to an interlocking pattern after fusion”: what is "their" referring to?

“their” referred to A/P interfaces. We have changed this sentence now.

- Subsection “Spatiotemporal regulation of tension and Bazooka at fusing interfaces”: missing a period.

We have now added this.

- Subsection “Spatiotemporal regulation of tension and Bazooka at fusing interfaces”: since supplementary figures are possible in eLife, why are these data not shown?

We have now included this data in Figure 2—figure supplement 1E-F.

- Subsection “Spatiotemporal regulation of tension and Bazooka at fusing interfaces”: why is this finding surprising?

We have removed the word “surprising”. The finding is unexpected and has not been previously described. As we now speculate in our Discussion section, the segment compartment specific differences in Vinculin and Bazooka levels may underlie the compartment specific contributions to fusion fidelity.

- Subsection “Cytoskeletal tension entrains the evolution of precision and modulates cell behaviour and interface geometry”: which parameters?

We have now detailed the parameters we have examined in these embryos.

- Subsection “Cytoskeletal tension entrains the evolution of precision and modulates cell behaviour and interface geometry”: should be "were" instead of "was".

We have corrected this.

- Subsection “Cytoskeletal tension entrains the evolution of precision and modulates cell behaviour and interface geometry”: what do the authors mean by "evolution"?

We have used evolution to indicate the temporal progression to high fusion fidelity. We wished to convey that fidelity is not pre-set in fusing segment compartments at the outset. We have now prefixed “evolution” with “temporal” to indicate this.

- Results section, Discussion section: some letters are missing.

Perhaps the letter Δ is missing because the fonts are different. I cannot find any letters missing.

- Subsection “Bazooka modulates interface remodeling and DME cell dynamics during fusion”: whereas, not "where as".

We have corrected this.